# IF1 promotes oligomeric assemblies of sluggish ATP synthase and outlines the heterogeneity of the mitochondrial membrane potential

Inés Romero-Carramiñana [1,2,3,4], Pau B. Esparza-Moltó [1,2,3,4], Sonia Domínguez-Zorita [1,2,3], Cristina Nuevo-Tapioles [1,2,3] & José M. Cuezva [1,2,3✉]

The coexistence of two pools of ATP synthase in mitochondria has been largely neglected despite in vitro indications for the existence of reversible active/inactive state transitions in the F1-domain of the enzyme. Herein, using cells and mitochondria from mouse tissues, we demonstrate the existence in vivo of two pools of ATP synthase: one active, the other IF1-bound inactive. IF1 is required for oligomerization and inactivation of ATP synthase and for proper cristae formation. Immunoelectron microscopy shows the co-distribution of IF1 and ATP synthase, placing the inactive "sluggish" ATP synthase preferentially at cristae tips. The intramitochondrial distribution of IF1 correlates with cristae microdomains of high membrane potential, partially explaining its heterogeneous distribution. These findings support that IF1 is the in vivo regulator of the active/inactive state transitions of the ATP synthase and suggest that local regulation of IF1-ATP synthase interactions is essential to activate the sluggish ATP synthase.

[1] Departamento de Biología Molecular, Centro de Biología Molecular Severo Ochoa, Consejo Superior de Investigaciones Científicas-Universidad Autónoma de Madrid (CSIC-UAM), 28049 Madrid, Spain. [2] Centro de Investigación Biomédica en Red de Enfermedades Raras (CIBERER) ISCIII, Madrid, Spain. [3] Instituto de Investigación Hospital 12 de Octubre, Universidad Autónoma de Madrid, Madrid, Spain. [4] These authors contributed equally: Inés Romero-Carramiñana, Pau B. Esparza-Moltó. ✉email: jmcuezva@cbm.csic.es

The mitochondrial ATP synthase is the rotatory engine placed in the inner membrane that uses the proton electrochemical gradient generated by the respiratory chain for the synthesis of cellular ATP in oxidative phosphorylation (OXPHOS)[1]. The ATP synthase is also an essential structural component for shaping mitochondrial cristae[2], participates in permeability transition (mPT) of the inner membrane to execute cell death[3,4] and forms part of the mitochondrial hub that controls intracellular signaling to allow cellular adaptation to changing stimuli[5–9]. Conformational changes in ATP synthase structure are the bases of these activities. In this regard, the activation and inhibition of ATP synthase, and of its $F_1$-ATPase domain, by similar anions and agents led Moyle and Mitchell to conclude that such effects were attributable to sluggish, reversible active/inactive state transitions in the $F_1$-domain of the enzyme[10].

The ATP synthase has a small inhibitory protein, called ATPase Inhibitory Factor 1 (IF1)[11], which is encoded in the nuclear *ATP5IF1* gene[12] and exerts its inhibitory activity by binding to the catalytic interface in the $F_1$ domain of the enzyme[13]. It is generally accepted that IF1 is an inhibitor only of the hydrolase activity of the ATP synthase[14]. Around neutrality, IF1 is thought to self-assemble into active dimers and, under alkaline conditions, into inactive tetramers and higher oligomers[15]. It has been suggested that in mitochondria, the oligomeric state of IF1 and its inhibitory potency are modulated by a combination of pH and cation-type effects, including $Ca^{2+}$ [15].

Highly proliferative cancer cells[16,17], undifferentiated mesenchymal stem cells[18] and stemness factor-mediated reprogrammed somatic cells[19] show high expression levels of IF1. In contrast, very large differences exist in the expression level of IF1 in mitochondria of differentiated human and mouse cells[20]. Studies in cells[16,17,21,22] and in isolated mitochondria of conditional tissue-specific mouse models of loss and gain-of-function of IF1[6,23–27], have shown that under basal conditions the mitochondrial content of IF1 inhibits both the synthetic and hydrolytic activities of the enzyme. These results contrast the findings in vitro supporting that IF1 is an inhibitor only of the hydrolase activity of the enzyme[14,15], and suggest that IF1 could represent the in vivo brake of the sluggish ATP synthase. Moreover, and in addition to the regulation of IF1 activity by pH, we have shown that PKA phosphorylation of S39 in IF1 prevents its binding and the inhibition of ATP synthase[28]. This mechanism of regulation is of relevance to increase the output of ATP in the heart in vivo upon an increase in energy demand[28], as well as to prevent ATP exhaustion in ischemia[29]. Overall, these findings support the idea that in some human and mouse tissues two pools of ATP synthase coexist in coupled mitochondria, one active and the other IF1-bound inactive.

Additionally, IF1 plays a relevant role in oligomerization of the ATP synthase and hence in mitochondrial cristae structure since its overexpression in cancer cells[30], and in vivo in hepatocytes[24], myocytes[26], cardiomyocytes[27] and neurons[6], promotes an increase in oligomeric assemblies of the ATP synthase. Contrariwise, its ablation in neurons and cardiomyocytes reduces the content of ATP synthase oligomers[6,27] resulting in altered mitochondrial cristae structure[6]. Remarkably, recent cryo-EM structures of mammalian ATP synthases have provided further support to the relevance of IF1 in enzyme oligomerization and to the existence of a fraction of IF1-inhibited ATP synthase in mitochondria[31,32]. Indeed, cryo-EM structures reveal that dimers of IF1 link two adjacent antiparallel dimers of ATP synthases to form inactive tetramers of the enzyme[31,32]. Herein, we have investigated the role played by IF1 in promoting oligomeric assemblies of inactive (sluggish) ATP synthase in vivo in different mouse tissues under basal physiological conditions of mitochondria.

## Results

**HCT116 and Jurkat cancer cells have a fraction of inactive ATP synthase.** HCT116 cells were used for the generation of IF1-knock-out (IF1-KO) cells by CRISPR-Cas9 technology. Subcellular fractionation (Fig. 1a) and immunofluorescence microscopy (Fig. 1b) confirmed ablation of IF1 in knockout cells. Ablation of IF1 had no effect on the rates of cellular proliferation (Supplementary Fig. 1a). The polyclonal anti-IF1 antibody developed in this study was further validated by assessing the colocalization of IF1 with mitochondrial β-F1-ATPase (Supplementary Fig. 1b) (Pearson's r = 0.90 ± 0.01). Ablation of IF1 promoted an increase in oligomycin-sensitive rate of ATP synthesis in permeabilized cells, as assessed following the kinetics of the luminescence production of ATP (Fig. 1c), and in the ATP hydrolytic activity in isolated mitochondria (Fig. 1d), supporting the idea that a fraction of mitochondrial ATP synthase is inhibited by its binding to endogenous IF1 under basal cellular conditions of cancer cells. Differences in ATP synthase activities between CRL and IF-KO cells were obliterated in the presence of oligomycin, indicating that both cell lines were equally sensitive to oligomycin (Fig. 1c, d). Similarly, the generation of Jurkat IF1-knockout cells (Supplementary Fig. 1c) also supported that IF1 is inhibiting a fraction of mitochondrial ATP synthase under basal cellular conditions (Supplementary Fig. 1d, e). Interestingly, determination of ATPase activity in clear native (CN)-gels of mitochondrial proteins from HCT116 cells further confirmed that IF1-KO cells had a higher activity of the enzyme than CRL cells (Fig. 1e).

Proximity Ligation Assays (PLA) using antibodies against human IF1 (this study) and β-F1-ATPase[33] supported the existence of IF1/β-F1-ATPase complexes in control HCT116 but not in IF1-KO cells (Fig. 1f). Similar results were obtained in Jurkat cells (Supplementary Fig. 1f). PLA data using antibodies against ATP synthase and/or IF1 have been presented as spots/area[34], spots/nucleus[35] or as percentage of spots/μm$^3$ of mitochondria using a projection of PLA signals onto mitochondrial immunostaining of the cells with TOMM20[36]. Using Mitotracker red staining, we found no significant differences in the mitochondrial volume between CRL (n = 10) and IF1-KO (n = 7) cells (348 ± 37 μm$^3$/cell versus 341 ± 24 μm$^3$/cell, respectively), supporting that mitochondrial mass is not affected by ablation of IF1. However, for calculation of PLA data, Mitotracker staining was omitted because it reduced the sensitivity of the assay as revealed by the drop in the number of PLA signals obtained when compared to non-stained cells (Fig. 1f).

Immunocapture of ATP synthase from HCT116 cells promoted the co-immunoprecipitation of a fraction of IF1 bound to β-F1-ATPase only in control cells (Fig. 1g), further supporting that mitochondria contain under basal conditions a fraction of IF1-bound inactive ATP synthase.

**Genetic ablation of IF1 affects mitochondrial structure and activities.** The basal and oligomycin sensitive respiratory (OSR) rates of IF1-KO cells were significantly upregulated when compared to their respective parental HCT116 (Fig. 2a) and Jurkat (Supplementary Fig. 1g) cells. However, the cellular rates of lactate production were not affected (Fig. 2b and Supplementary Fig. 1h). Both ΔΨm and mitochondrial ROS (mtROS) production vary as a function of IF1 dose[6]. In agreement with this observation, ΔΨm and mtROS production increased in IF1 expressing cells when compared to IF1-KO cells (Fig. 2c, d). To rule out the implication of multidrug resistance (MDR) pumps in differences in TMRM staining between the two genotypes, we treated cells with cyclosporin H (CsH) and found no relevant differences

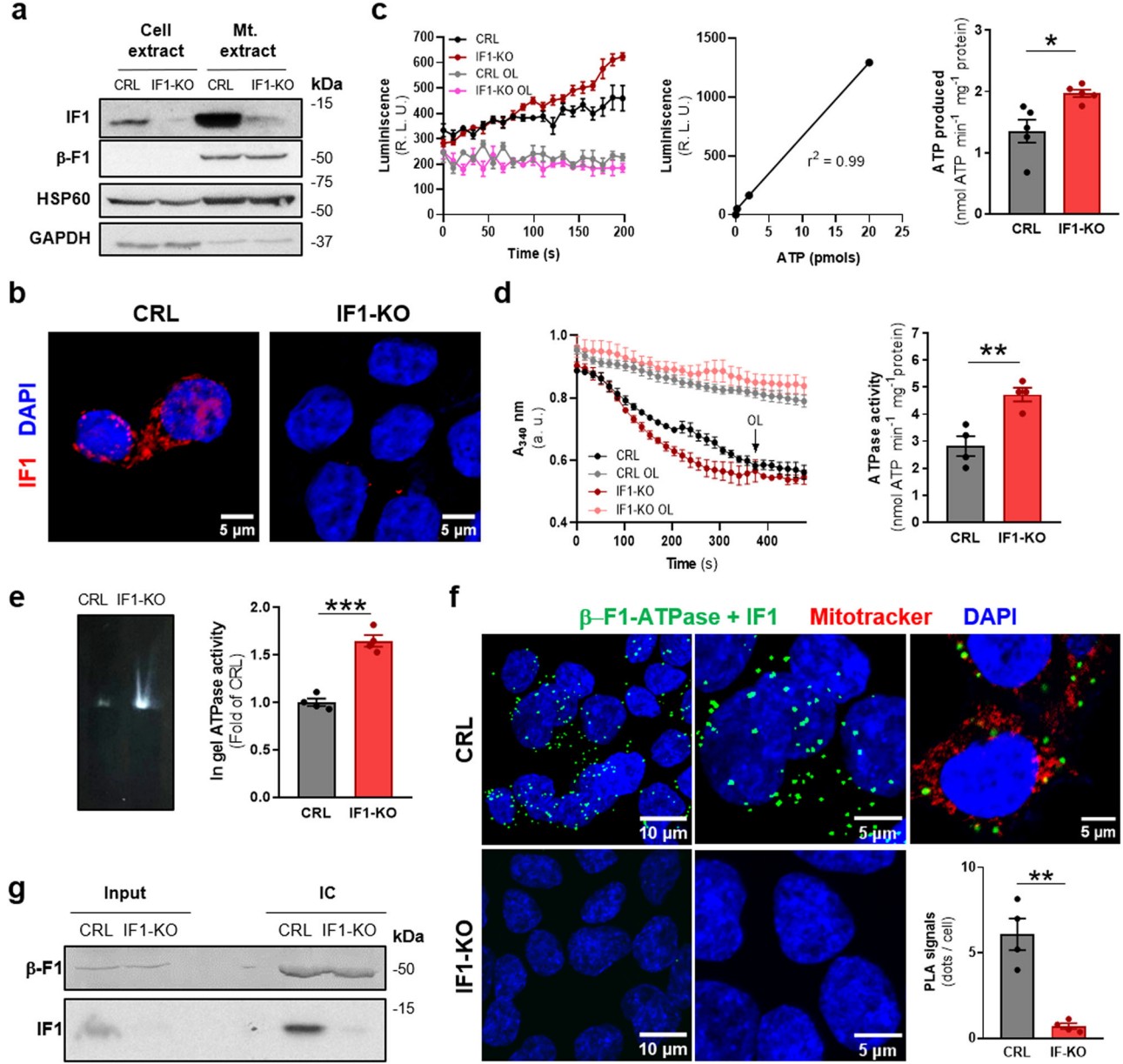

**Fig. 1 HCT116 cells contain a fraction of inactive IF1-bound ATP synthase under basal conditions. a** Representative blots of IF1 and β-F1-ATPase expression in cellular and mitochondrial (Mt.) extracts of CRL and IF1-KO HCT116 cells. The expression of mitochondrial HSP60 and cytosolic GAPDH are shown as loading controls. Molecular weight markers are indicated to the right of the blots. **b** Representative immunofluorescences of IF1 expression (red) in CRL and IF1-KO HCT116 cells. DAPI (blue) stained nuclei. **c** Kinetic representation of the synthetic activity of ATP synthase in digitonin-permeabilized CRL and IF1-KO HCT116 cells. 2 μM Oligomycin (OL) was used to inhibit ATP synthase activity ($n = 5$). ATP curve used to convert the luminescence (Relative Light Units, R. L. U.) into nmoles of ATP produced. Histograms show the rate of ATP production per mg of cellular protein in CRL and IF1-KO HCT116 cells. **d** Kinetic representation and histograms showing the hydrolytic activity of ATP synthase in isolated mitochondria of CRL and IF1-KO HCT116 cells. 2 μM Oligomycin (OL) was added to inhibit the hydrolytic activity of the ATP synthase either at the beginning or at the indicated time point ($n = 4$). **e** Representative ATP hydrolytic activity of complex V in CN-PAGE gels of mitochondria from CRL and IF1-KO HCT116 cells. The histograms show the quantification of the in-gel activity ($n = 4$). **f** Representative images of Proximity Ligation Assays (PLA) showing the interaction between β-F1-ATPase and IF1 (green dots) in CRL and IF1-KO HCT116 cells. DAPI (blue) stained nuclei. The right image shows the localization of PLA dots onto a single plane of mitochondria stained with Mitotracker Red (red) in CRL HCT116 cells. Histograms show the number of PLA signals per cell in CRL ($n = 4$) and IF1-KO ($n = 4$) HCT116 cells. **g** Immunocapture (IC) of ATP synthase from mitochondrial extracts of CRL and IF1-KO HCT116 cells blotted against β-F1-ATPase (β-F1) and IF1. Input, 20 μg of mitochondrial extract. "n" indicates the number of independent experiments. The plots and histograms show the mean and the error bars ±SEM. *$p \leq 0.05$; **$p \leq 0.01$; ***$p \leq 0.001$ when compared to CRL by Student's *t* test. See also Supplementary Fig. 1.

when compared to values without CsH (Fig. 2c). Oligomycin treatment obliterated the differences in ΔΨm and mtROS production between the two genotypes (Fig. 2c, d), supporting that these differences are due to an enhanced ATP synthase activity in IF1-KO cells. Hence, these results support the idea that IF1 participates in the control of basic mitochondrial functions such as respiration, ΔΨm and mtROS production.

Determination of the activities of complex I, II and IV of the respiratory chain in isolated mitochondria revealed no relevant differences between the two cellular genotypes (Supplementary

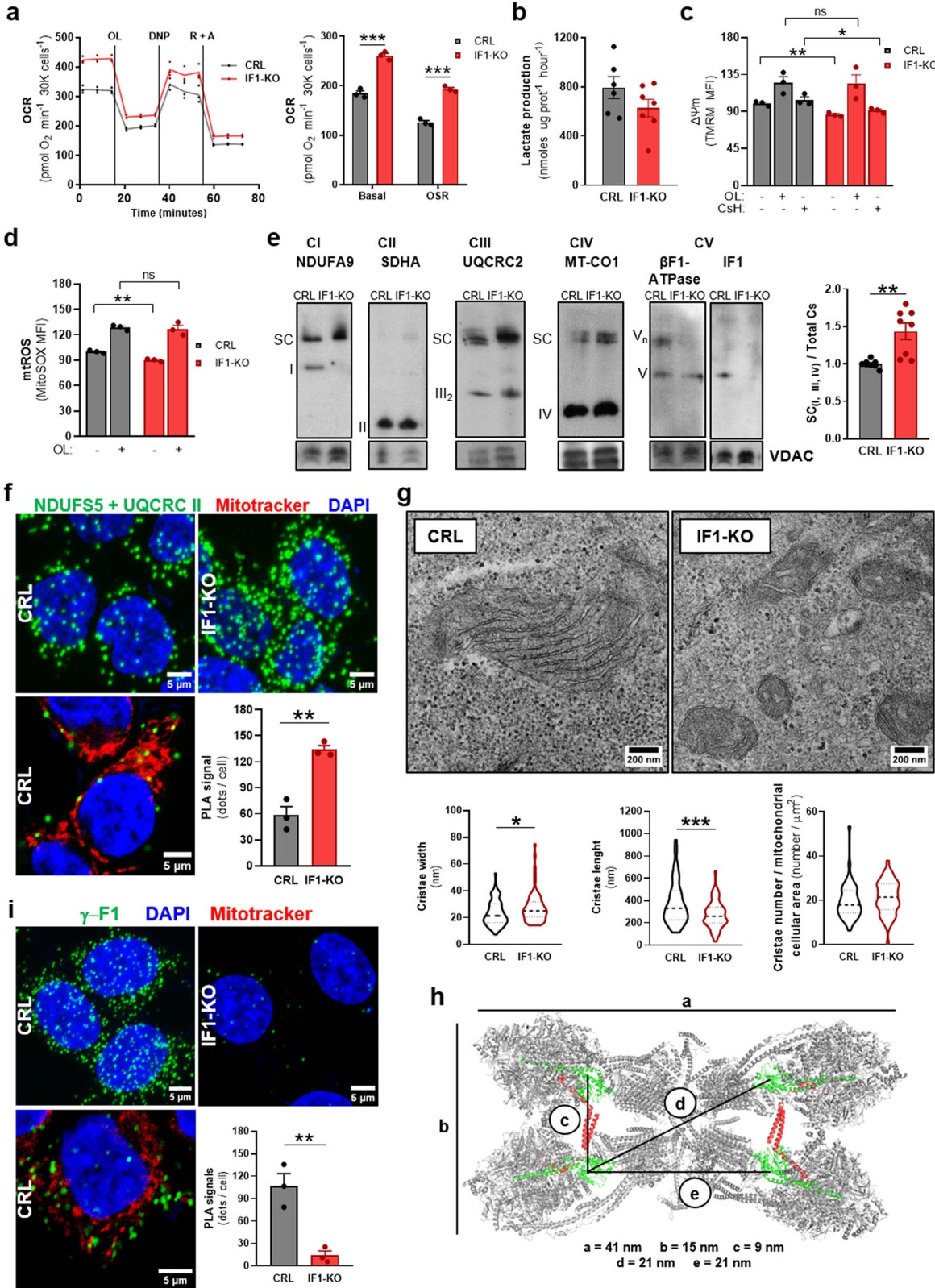

Fig. 2a). Consistently, there was lack of relevant changes in the expression of subunits of respiratory complexes (Supplementary Fig. 2b). The enhanced rates of basal cellular respiration in IF1-KO cells (Fig. 2a) could result from an increased assembly of respiratory complexes in the supercomplex (SC) containing complexes I, III and IV. Indeed, mitochondrial membranes of IF1-KO cells showed increased SC assembly when compared to control cells in BN-gels (SC in Fig. 2e). In fact, free complex I almost vanished in IF1-KO cells to show a sharp increase in SC (Fig. 2e). Interestingly, HCT116 cells showed very little content of complex III2/IV (Fig. 2e), in agreement with previous findings[22].

To provide additional evidence of an increase in SC formation in IF1-KO cells (Fig. 2e), that could explain their enhanced rates of respiration (Fig. 2a), we decided to develop an approach based

**Fig. 2 Ablation of IF1 prevents oligomerization of ATP synthase. a** Representative profile of oxygen consumption rates (OCR) of control (CRL) and IF1-KO HCT116 cells ($n = 3$) with glucose as respiratory substrate. The addition of oligomycin (OL), 2,4-dinitrophenol (DNP) and rotenone (R) plus antimycin A (A) is indicated. The histograms show the quantification of the mitochondrial basal and oligomycin sensitive respiration (OSR) ($n = 3$). **b** Glycolytic activity measured by the rate of lactate production in CRL ($n = 6$) and IF1-KO ($n = 7$) HCT116 cells. **c** Mitochondrial membrane potential ($\Delta\Psi$m) in CRL ($n = 3$) and IF1-KO ($n = 3$) HCT116 cells treated or not with OL or cyclosporin H (CsH). MFI, mean fluorescent intensity. ns, no significant. **d** Mitochondrial reactive oxygen species (mtROS) production in CRL and IF1-KO HCT116 cells untreated ($n = 3$) or treated ($n = 3$) with OL. **e** Representative BN-immunoblots of oxidative phosphorylation complexes. From left to right: Complex I (NDUFA9), II (SDHA), III (UQCRC2), IV (MT-COI) and V ($\beta$-F1-ATPase and IF1) in isolated mitochondria from CRL and IF1-KO HCT116 cells. The migration of supercomplexes (SC) and oligomers of complex V (Vn) is also indicated. VDAC is shown as loading control. Histograms show the relative increase in SC in IF1-KO cells, at least, three independent experiments were carried out. **f** Representative images of Proximity Ligation Assays (PLA) showing the interaction (green dots) between Complex I (NDUFS5) and Complex III (UQCRC2) in CRL and IF1-KO HCT116 cells. DAPI (blue) stained nuclei. The image to the bottom shows the localization of PLA dots onto a single plane of mitochondria stained with Mitotracker Red (red) in CRL HCT116 cells. The histograms show the number of PLA signals per cell ($n = 3$). **g** Representative electron micrographs of mitochondria from CRL and IF1-KO cells. Violin plots show the quantification of cristae width ($n = 73$–$74$), cristae length ($n = 76$–$91$) and cristae number per mitochondrial cellular area ($n = 45$-$47$). **h** Cryo-EM structure of porcine ATP synthase tetramer viewed from the matrix side. IF1 is highlighted in red and $\gamma$-F1-ATPase in green. a-e lines indicate ~ distance in nm. Molecular reconstruction from PDB 6J5K. Image created with PyMOL Molecular Graphics System. **i** Representative images of PLA using $\gamma$-F1-ATPase as target (green dots) in CRL and IF1-KO HCT116 cells. DAPI (blue) stained nuclei. The lower image shows the localization of PLA dots onto a single plane of mitochondria stained with Mitotracker Red (red) in CRL HCT116 cells. Histograms show the number of PLA signals per cell ($n = 3$). "n" indicates the number of independent experiments. The histograms show the mean and the error bars ± SEM. *$p \leq 0.05$; **$p \leq 0.01$; ***$p \leq 0.001$ when compared to CRL by t-Student's test. See also Supplementary Fig. 2.

on PLA. Immunofluorescence of HCT116 cells with anti-NDUFS5, a subunit of Complex I, and anti-UQCRCII, a subunit of CIII, specifically decorated the mitochondrial cellular network (Pearson's $r = 0.88 \pm 0.01$ for CRL and $0.90 \pm 0.01$ for IF1-KO cells) (Supplementary Fig. 2c), validating the use of these antibodies in PLA assays. Moreover, UQCRCII immunostaining confirm no differences in mitochondrial area between CRL ($n = 37$) and IF1-KO ($n = 34$) cells ($79 \pm 2\,\mu m^2$ versus $78 \pm 2\,\mu m^2$, respectively). PLA assays using the former antibodies confirmed that IF1-KO cells had increased SC assembly when compared to control cells (Fig. 2f), further supporting that the enhanced respiration of IF1-KO cells results from a facilitated SC formation.

**IF1 is required to generate oligomers of ATP synthase.** IF1 is involved in the generation of oligomeric structures of ATP synthase in vivo[6] that might result in the tetrameric structures of inactive ATP synthase[27,31,32]. Consistently, ablation of IF1 in HCT116 cells promoted the disassembly of oligomeric structures of ATP synthase as assessed in BN-gels (Fig. 2e). Oligomerization of the ATP synthase is required for the correct assembly of mitochondrial cristae[2,37,38]. Electron microscopy (EM) analysis revealed that mitochondrial structure was significantly affected by ablation of IF1 (Fig. 2g). In fact, mitochondria in IF1-KO cells showed reduced area and enhanced circularity than in control cells (Supplementary Fig. 2d). Moreover, cristae structure was also affected as showed by the wider and shorter cristae of mitochondria devoid of IF1 (Fig. 2g). In fact, IF1-KO cells showed reduced number of cristae per mitochondrion ($5.2 \pm 0.6$) when compared to control ($8.8 \pm 0.7$, $p = 0.0002$). However, when the number of cristae was normalized by the mitochondrial area of the cell, the content of cristae between the two cellular genotypes was the same (Fig. 2g). Consistent with the overall lack of differences in cristae number, the cellular content of OPA1 and MIC60, which are also involved in cristae formation, was the same in CRL and IF1-KO cells (Supplementary Fig. 2e), supporting that total mitochondrial mass of the cell is similar between the two genotypes. Overall, these results support that IF1 plays a relevant structural role in oligomerization of ATP synthase and in cristae structure.

The ATP synthase only contains one $\gamma$-subunit per monomer of enzyme[2,14]. Hence, to visualize oligomers of ATP synthase in cells we carried out PLA assays using the $\gamma$-F1-ATPase subunit of the enzyme as target of the antibody to anchor the primers of

DNA polymerase. The anti-$\gamma$ antibody used specifically recognized the mitochondrial protein both in blots (Supplementary Fig. 2b) and in immunofluorescence (Supplementary Fig. 2f). $\gamma$-subunits in adjacent or in opposite dimers of tetrameric assemblies of the ATP synthase are at ~8 or 20/21 nm distance from other $\gamma$-subunits (Fig. 2h)[31,32], which are distances lower than the 40 nm limit for positive signals in PLA assays. The results obtained confirmed the detection of oligomers of ATP synthase within the mitochondrial reticulum in IF1-expressing HCT116 (Fig. 2i) and Jurkat cells (Supplementary Fig. 2g). Moreover, PLA signals were significantly reduced in IF1-KO cells (Fig. 2i and Supplementary Fig. 2g), strongly supporting that IF1 is indispensable for the assembly of ATP synthase oligomers.

**The mitochondrial distribution of IF1 determines $\Delta\Psi$m heterogeneity.** $\Delta\Psi$m is the main driver of OXPHOS and it is not homogenously distributed along mitochondria[39,40]. Since the binding of IF1 to the ATP synthase prevents the backflow of $H^+$ into the matrix and contributes to mitochondrial hyperpolarization[16,17], we speculated that intramitochondrial IF1 distribution could contribute to the heterogeneity of $\Delta\Psi$m[39,40] and perhaps of mtROS production. To this aim, we developed HCT116 cell lines stably expressing IF1-GFP fusion protein (Fig. 3a) and a version of the same construct (IF1-S39E-GFP) (Fig. 3b) that contains a mutation in IF1 that prevents its binding to ATP synthase[28], to study by STED-super resolution microscopy the co-distribution of TMRM and GFP fluorescence peaks, the former a probe of $\Delta\Psi$m. Subcellular fractionation (Fig. 3a, b) and immunofluorescence microscopy (Supplementary Fig. 3a) confirmed that the fusion proteins were efficiently targeted to mitochondria (Pearson's $r = 0.86 \pm 0.007$ for IF1-GFP and $0.85 \pm 0.01$ for IF1-S39E-GFP). As an additional control of the study, we used GFP targeted to mitochondria by the presequence of $\beta$-F1-ATPase (p$\beta$-GFP)[41], which is also efficiently targeted to mitochondria of HCT116 cells as revealed by subcellular fractionation (Fig. 3c) and immunofluorescence microscopy (Supplementary Fig. 3a) (Pearson's $r = 0.86 \pm 0.001$). As expected, only the IF1-GFP construct was able to interact with ATP synthase as revealed by co-immunoprecipitation of GFP with the enzyme (Fig. 3d).

Interestingly, STED microscopy with any of the GFP constructs studied revealed homogenous but discrete domains inside mitochondria, which correspond to different cristae (Fig. 3e). However, the distribution of TMRM varied greatly between

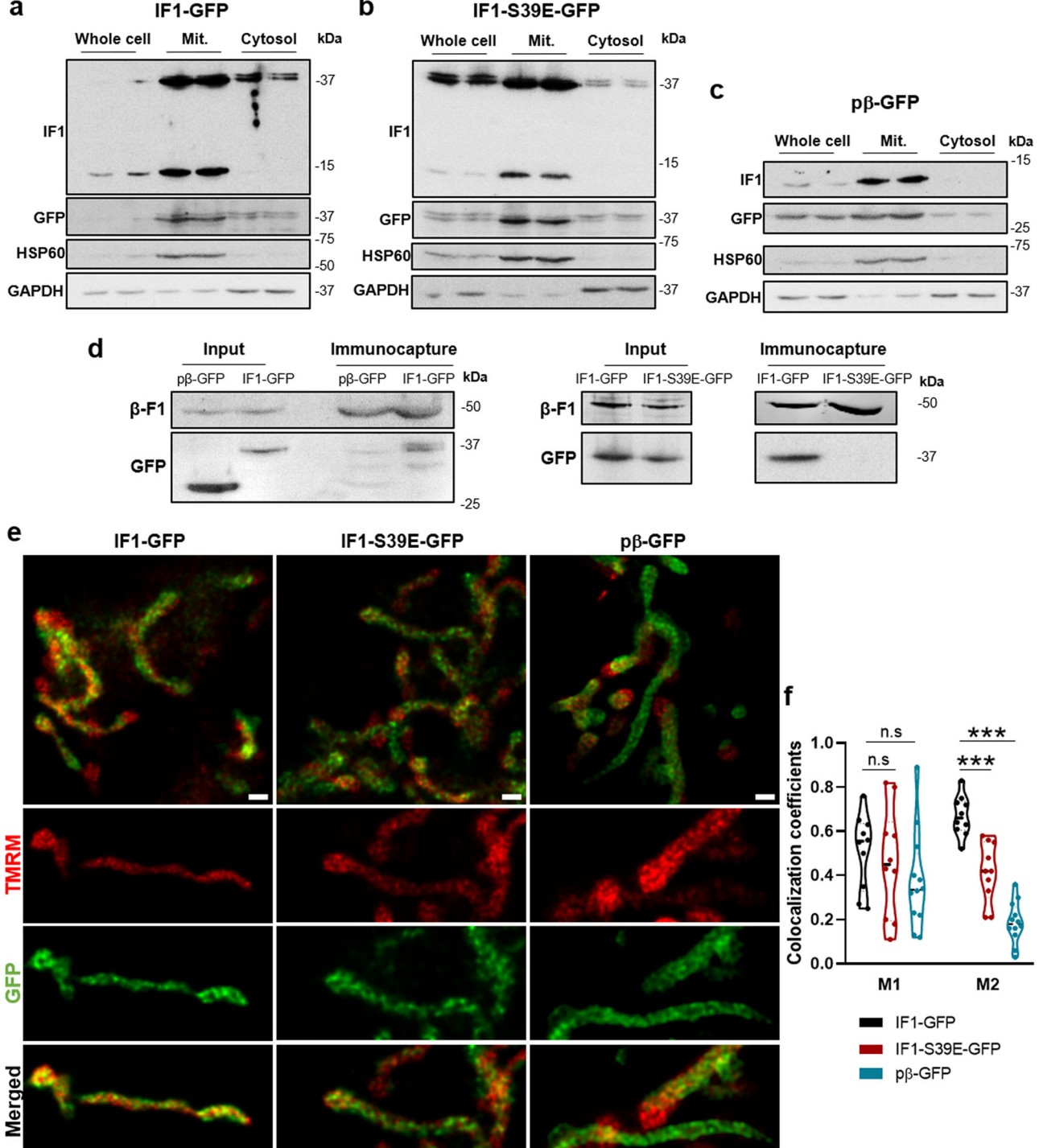

**Fig. 3 IF1 contributes to define the heterogeneity of mitochondrial membrane potential.** Representative blots of IF1 and GFP expression in cellular, mitochondrial (Mit.) and cytosolic extracts of HCT116 cells expressing IF1-GFP (**a**), IF1-S39E-GFP (**b**) and pβ-GFP constructs (**c**). The expression of mitochondrial HSP60 and cytosolic GAPDH are also shown as loading controls. Molecular weight markers are indicated to the right of the blots. **d** Immunocapture of ATP synthase from mitochondrial extracts of pβ-GFP, IF1-GFP and IF1-S39E-GFP HCT116 cells blotted against β-F1-ATPase (β-F1) and GFP. Input, 20 µg of mitochondrial extract. **e** Representative high-resolution images of mitochondria from IF1-GFP, IF1-S39E-GFP and pβ-GFP HCT116 live cells stained with TMRM (red). Three independent experiments were carried out. Scale bars 0.5 µm. **f** Violin plots show the Manders' colocalization coefficients of TMRM and GFP fluorescence pixel by pixel of the images ($n = 10$–12 fields of three independent experiments). Images were aligned and deconvoluted via Hyugens software and analyzed with the JACoP plugin of ImageJ. M1 represents the proportion of high TMRM signals that overlap with GFP signals and M2 the proportion of GFP signals that overlap with high TMRM signals. In both TMRM and GFP fluorescence, the threshold used only selected the pixels with maximal fluorescence intensity of GFP and TMRM. n.s., no significant. **$p \leq 0.01$; ***$p \leq 0.001$ when compared by one-way ANOVA and Dunnett's multiple comparison tests. See also Supplementary Fig. 3.

cristae throughout mitochondria (Fig. 3e), in agreement with previous findings[39], supporting the idea that cristae represent functional independent units.

Analysis of the co-distribution of TMRM maximal intensity with mitochondrial GFP intensity revealed significant differences between the three cell lines (Fig. 3e, f). In fact, whereas peaks of TMRM fluorescence largely coincided with peaks of the distribution of IF1-GFP (Fig. 3e, f) that was not the case in IF1-S39E-GFP or in GFP expressing cells (Fig. 3e, f), as revealed in Manders M2 colocalization coefficients (Fig. 3f), that analyzed the proportion of GFP signals that overlap with maximum TMRM signals. These results support that part of the heterogeneity of ΔΨm is linked to the intramitochondrial distribution of IF1-bound to the ATP synthase. Interestingly, it appears that the IF1-inhibited enzyme is preferentially localized towards cristae placed at the apical regions of the mitochondrion (Fig. 3e).

The partial arrest of the respiratory chain by inhibition of the ATP synthase forces the production of mtROS at complexes of the electron transport chain[17,42]. Next, we studied by STED-super resolution microscopy the co-distribution of MitoSOX, a probe to assess mitochondrial production of superoxide radical, with IF1-GFP or GFP fluorescence. In contrasts to TMRM data, the variation of MitoSOX fluorescence in mitochondria was not significantly different between IF1-GFP and GFP expressing cells (Supplementary Fig. 3b, c), perhaps because of different intramitochondrial localization of the probes and the short half-life and high diffusion rate of superoxide radical. Overall, the results in cells support the idea that under basal conditions there is a fraction of IF1-bound and inhibited ATP synthase that is responsible for promoting enzyme oligomerization that affects cristae structure and the heterogeneous distribution of ΔΨm.

**IF1 determines the oligomeric assemblies of sluggish ATP synthase in mouse tissues.** We have developed a global-tissue $IF1^{KO}$ mice, by deleting exon 3 of the $Atp5if1$-floxed mouse[6], through Cre-mediated recombinase expression in the male germ line (Fig. 4a), to investigate the relevance of IF1 in vivo. In agreement with the tissue-restricted expression of IF1 in mouse tissues[20], the global $IF1^{KO}$ mice showed no IF1 expression in brain, kidney and heart, confirming targeting of $Atp5if1$ (Fig. 4b). As reported in different mouse strains[20], the liver and skeletal muscle are tissues that do not express IF1 protein (Fig. 4b). Global $IF1^{KO}$ mice showed no gross phenotypic alterations when compared to floxed animals used as controls (Supplementary Fig. 4a), in agreement with findings in a previous global $IF1^{KO}$ mouse model[43].

Determination of the hydrolase activity of ATP synthase in isolated mitochondria of control and $IF1^{KO}$ mice confirmed that brain, kidney and heart contained a fraction of IF1-inhibited ATP synthase in their mitochondria (Fig. 4c). The fraction of inactive ATP synthase correlates with the amount of IF1 present in these tissues (brain>kidney>heart)[20]. Similar findings were obtained by the determination of the ATP synthetic activity of isolated mitochondria from brain, kidney and heart in the presence of succinate and ADP (Fig. 4d). On the contrary, tissues such as liver and skeletal muscle, which are devoid of IF1 (Fig. 4b)[20], have no fraction of inhibited enzyme, as assessed by the hydrolase (Fig. 4c) and synthase (Fig. 4d) activities of ATP synthase.

Conditional mouse models of loss and gain of function of IF1 in forebrain neurons have shown that the higher the mitochondrial dose of IF1 is, the higher is the formation of oligomeric assemblies of ATP synthase[6]. Herein, we have studied the oligomeric assemblies of ATP synthase in BN-gels of four additional mouse tissues representing high, medium, and negligible expression levels of IF1, which are respectively represented by brain, kidney, heart, liver and skeletal muscle (Fig. 4e)[20]. BN-gels revealed the existence of monomers and oligomers of ATP synthase in brain, kidney and heart (Fig. 4e). Consistent with data in human cells (Fig. 2e) and in mouse neurons[6], ablation of IF1 in these tissues prevented the formation of ATP synthase oligomers (Fig. 4e). Remarkably, mouse liver and skeletal muscle which expresses negligible levels of IF1 (Fig. 4b), showed no evidence of ATP synthase oligomers in preparations of control mice (Fig. 4e), in agreement with previous findings[24]. Remarkably, oligomers of ATP synthase only appeared in mouse liver[24] and in skeletal muscle[26] when the IF1-H49K mutant is overexpressed. Interestingly, most of IF1 co-fractionated with monomers and oligomers of the ATP synthase supporting IF1-ATP synthase interaction (Fig. 4e). In the case of brain, kidney and heart preparations there was also a relevant fraction of IF1 migrating at higher electrophoretic mobility corresponding to IF1-bound ATP synthase oligomers (Fig. 4e). CN-gels of brain, kidney and heart preparations confirmed that ablation of IF1 increased ATPase activity of the enzyme (Fig. 4f), further supporting the interaction of IF1 with the ATP synthase under basal conditions of mitochondria in the three tissues. Moreover, CN-gels also support that oligomers of the ATP synthase bound to IF1 are inhibited (Fig. 4f). As expected, liver and skeletal muscle of IF1-KO mice show no differences in the in-gel ATPase activity (Fig. 4f), nor the appearance of ATPase activity in oligomers (Fig. 4f) when compared to controls. However, it should be noted the appearance of a small fraction of oligomers in CN-gels of brain mitochondria of IF1-KO mice perhaps because of the self-assembly of ATP synthase monomers independent of IF1 (Fig. 4f). Altogether, these results support that IF1 is required for oligomerization of the ATP synthase in vivo and support that those tissues that express IF1, such as kidney and heart (Fig. 4b), as previously reported for brain[6], have a fraction of IF1-bound an inhibited enzyme under normal basal conditions of mitochondria.

Immunocapture of ATP synthase from isolated mitochondria of control and $IF1^{KO}$ mice further confirmed that tissues that express IF1 have a fraction of it bound to the ATP synthase (Fig. 4g), whereas that was not the case for liver and skeletal muscle (Fig. 4g) given the negligible expression of IF1. These results strongly support the existence of a fraction of IF1-inhibited ATP synthase in mitochondria under non-stress conditions in the mouse tissues that express IF1.

Interestingly, as previously reported[20], in blots of brain and kidney extracts (Fig. 4b) and in their isolated mitochondria (Fig. 4g), IF1 appears as a double band. Differences in electrophoretic migration of IF1 results from the phosphorylation or not of S39[28], as revealed in 2D-gels by the lower electrophoretic migration of the phospho-mimetic IF1-S39E mutant when compared to the phospho-deficient IF1-S39A mutant (Fig. 4h). Remarkably, in kidney and brain most of the IF1 immunocaptured with ATP synthase corresponds to the dephosphorylated protein, i.e., the band with higher electrophoretic mobility (Fig. 4h), which is the isoform of IF1 able to interact with the enzyme[28]. However, in heart mitochondria the immunocapture pulls down both phospho- and dephospho-IF1 (Fig. 4g), supporting that in this tissue there are additional phosphorylation sites in IF1 that do not impede its interaction with ATP synthase.

**Intramitochondrial distribution of ATP synthase and IF1 in mitochondria of mouse tissues.** As an approach to visualize the distribution of ATP synthase and IF1 in mitochondria of mouse tissues, we used high-resolution gold-immunolabeling of Lowicryl embedded tissue sections[44]. Figure 5a shows representative

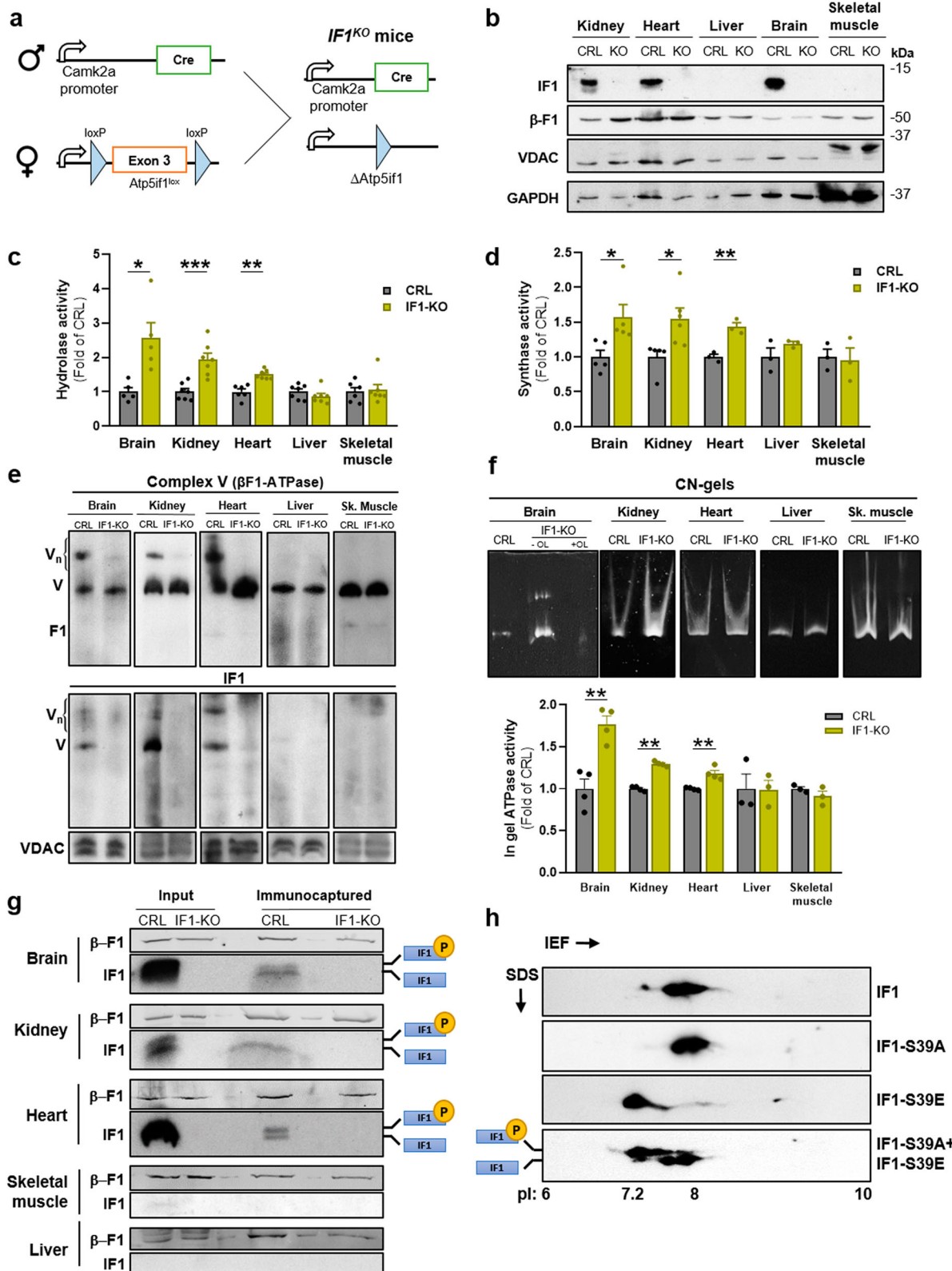

examples of gold decoration of mitochondria using antibodies against β-F1-ATPase or IF1 in mouse kidney and heart preparations. Consistent with blot data (Fig. 4b), the number of β-F1-ATPase particles/mitochondrial area in kidney and heart is quite similar (Fig. 5a), whereas IF1 labeling is significantly higher in kidney than in heart (Fig. 5a), paralleling the relative abundance of the proteins in each tissue (Fig. 4b)[20]. The distribution of

gold labeling in different cristae structures (Supplementary Fig. 4b) indicated that most β-F1-ATPase and IF1 particles are localized in cristae sheets followed by the tips (Supplementary Fig. 4c). However, the abundance of β-F1-ATPase particles/μm of crista tips, considering 30–40 nm of crista at either side of the tip, was much higher than in cristae sheets (Fig. 5b), in agreement with the preferential cristae localization described for the ATP

**Fig. 4 Genetic ablation of IF1 prevents oligomerization and sluggish ATP synthase in mouse tissues. a** Scheme of the generation of IF1-KO mice in the male germ line. **b** Representative blot of IF1 and β-F1-ATPase (β-F1) expression in tissue extracts of kidney, heart, liver, brain and skeletal muscle of CRL and IF1-KO mice. VDAC and GAPDH are shown as loading controls. Molecular weight markers are indicated to the right of the blots. Hydrolase (**c**) and synthase (**d**) activities of ATP synthase in isolated mitochondria from brain, kidney, heart, liver and skeletal muscle of CRL and IF1-KO mice ($n = 5–7$). **e** Representative BN-immunoblots of complex V in brain, kidney, heart, liver and skeletal (sk.) muscle mitochondria from CRL and IF1-KO mice blotted against β-F1-ATPase and IF1 of at least three independent experiments. The migration of the monomeric (V) and oligomeric (Vn) complex V is indicated. VDAC is shown as loading control. **f** Representative ATP hydrolytic activity of complex V in CN-gels of mitochondria from brain, kidney, heart, liver and skeletal muscle of CRL and IF1-KO mice of at least three independent experiments. Where indicated 2 μM oligomycin (OL) was added in the sample to inhibit ATP hydrolysis. The histograms show the quantification of the in-gel activity ($n = 4$). **g** Representative ATP synthase immunocaptured from isolated mitochondria of brain, kidney, heart, skeletal muscle and liver of CRL and IF1-KO mice blotted against β-F1-ATPase (β-F1) and IF1. Input, 20 μg of mitochondrial extract. The migration of the phospho- and dephospho-IF1 isoforms is indicated. **h** Representative blots of 2D-gels showing the migration of phosphodeficient S39A and phosphomimetic S39E IF1-mutants expressed in NRK cells. pI: Isoelectric point. "n" refers to the number of independent mice used. The histograms show the mean and the error bars ± SEM. *$p \leq 0.05$; **$p \leq 0.01$; ***$p \leq 0.001$ when compared to CRL by Student's $t$ test. See also Supplementary Fig. 4.

synthase[2]. However, it should be taken into consideration that ATP synthase labeling at cristae sheets reveals both ATP synthase at the sheets and at the rims of the sheets[38,45]. Hence, the 3-fold higher estimation of ATP synthase at cristae tips (Fig. 5b) is even an underestimation of ATP synthase at cristae rims, and supports that ATP synthase at the sheets represents a small fraction of the available enzyme. The labeling of cristae junctions and inner boundary membranes (Supplementary Fig. 4c) was negligible like the background of the technique ($2.4 \pm 0.3$ and $0.9 \pm 0.2$ gold particles/μm$^2$ of nuclear labeling for β-F1-ATPase and IF1, respectively).

Double immunolabeling of kidney, heart, brain, and liver sections using colloidal gold of 10 nm (β-F1-ATPase) and 15 nm (IF1) particles confirmed that the abundance of ATP synthase per square μm of mitochondrial section is similar in kidney, heart and brain and significantly higher than that in liver mitochondria (Fig. 5c, d). Immunolabeling of IF1 in kidney and brain mitochondria was the same and higher than that in heart (Fig. 5c, d). Consistent with blot data (Fig. 4b), IF1 labeling in liver was negligible (Fig. 5c, d). Interestingly, a significant percentage of β-F1-ATPase signals (~15%) appeared as doublets/triplets in the four tissues (Fig. 5e), suggesting decoration of ATP synthase oligomers. Moreover, a high percentage of IF1 signals were in the proximity (<15 nm) of β-F1-ATPase signals in the three tissues that express IF1 (Fig. 5f), also supporting the relevance of IF1 in oligomerization of ATP synthase in mouse tissues that express the inhibitor protein.

## Discussion

Herein, we show that ablation of IF1 in cells and mouse tissues that express IF1 promotes an increase in the ATP synthetic and hydrolytic activities of mitochondrial ATP synthase. In contrasts, tissues that naturally do not express IF1, such as skeletal muscle and liver, the ablation of IF1 has no effect in the activity of the enzyme. These results suggest that under basal conditions mitochondria that express IF1 contain a fraction of IF1-inhibited ATP synthase, in agreement with similar findings in brain mitochondria[6]. The finding of a fraction of IF1-inhibited enzyme is further supported by PLA assays in cells, the co-fractionation of IF1 with ATP synthase monomers and oligomers in BN-gels and by the co-immunoprecipitation of a fraction of dephosphorylated IF1 with ATP synthase in cells and in mouse tissues. Moreover, our findings are also in agreement with recent cryo-EM structures of isolated mammalian ATP synthases[31,32], in which dimers of IF1 join together two adjacent antiparallel dimers of inactive ATP synthases[27,46]. Overall, these findings encourage us to re-evaluate the widely held idea that IF1 only binds to the ATP synthase under mitochondrial de-energized conditions (hypoxia) to

prevent its hydrolase activity[14,47]. In contrasts, we support that the IF1-inhibited ATP synthase is the in vivo representative of the "sluggish" ATP synthase observed years ago in mitochondrial preparations[10]. As reported in that study[10], changes in enzyme activity were not attributable to effects on reaction velocity, but to reversible inactive/active state transitions in the F1 domain of the sluggish ATP synthase. We suggest that the regulated interaction of IF1 with the F1 domain of the enzyme promotes the active/inactive state transitions on the ATP synthase. The question now is, what is the physiological meaning of a fraction of inactive ATP synthase in respiratory tissues? We support that the sluggish ATP synthase represents a reservoir of enzyme that could become activated to supply ATP upon an increase in energy demand, as we have reported in heart mitochondria in vivo in response to β-adrenergic signaling[28]. Moreover, this reservoir of enzyme is also a relevant generator of mtROS by the ETC for signaling and adaptation of tissue responses[6,25].

Dimers of ATP synthase are essential for the bending of the lipid bilayer at cristae rims[48], and self-assemble into rows of dimers along cristae tips to provide the characteristic infolds of the inner mitochondrial membrane[2,49]. As assessed in BN-gels, CN-gels and PLA assays, we show that the expression of IF1 favors the formation of inactive oligomers of ATP synthase whereas IF1 ablation prevents formation of oligomers affecting the ultrastructure of cristae. IF1-driven changes in cristae structure are exerted without any alteration in OPA1 or MICOS expression, two other main players of mitochondrial cristae structure. Hence, these results support that IF1 is a critical subunit of the enzyme for its oligomerization and inhibition, in agreement with the role assigned to IF1 in cryo-EM structures of tetrameric ATP synthase[31,32] and in isobaric quantitative Protein Interaction Reporter (iqPIR) cross-linking studies[27] and hence, a relevant factor of cristae structure in some mammalian cells. However, there are tissues such as liver and skeletal muscle that are devoid of IF1 and lack an sluggish ATP synthase and oligomeric assemblies of the enzyme, but contain well developed cristae, suggesting that self-assembly[49] of ATP synthase is sufficient for correct cristae formation in these tissues.

ATP synthase and hydrolase activity assays in mitochondria of control and IF1-KO mice support that brain and kidney mitochondria contain a very large fraction of IF1-inhibited enzyme whereas in heart this proportion is less and none in liver or skeletal muscle. Consistently, immunoelectron microscopy data of β-F1-ATPase and IF1 content nicely correlated with activity data, supporting similar fractions of sluggish ATP synthase in these tissues. A relevant number of β-F1-ATPase immunoreactivity appeared as double/triple signals (aprox. 15%) and in proximity of IF1 signals (20-30%), what might suggest the recognition of IF1-inhibited ATP synthase oligomers. Similarly,

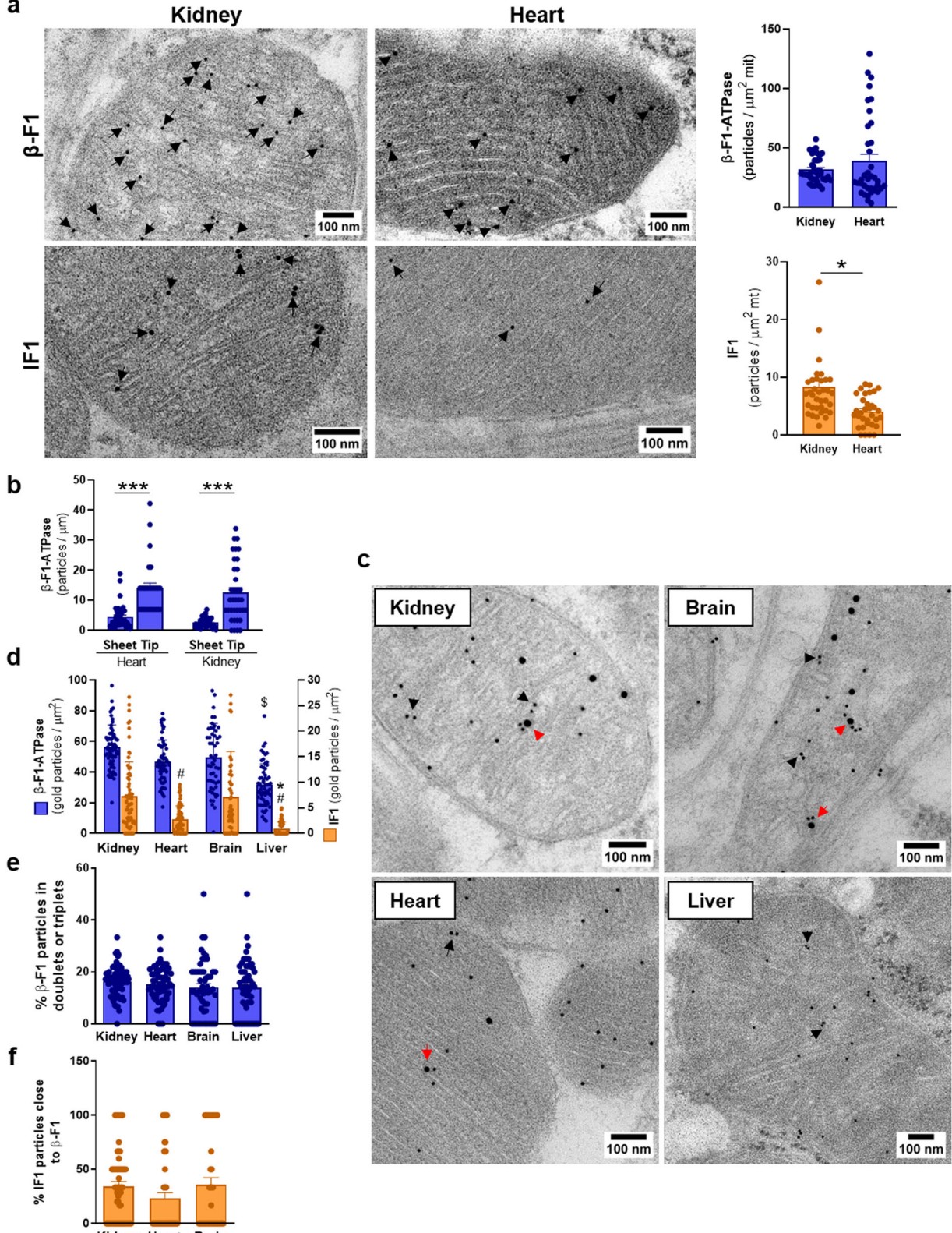

changes in enzyme activity in IF1-KO cells correlated with data from PLA signals of the γ-subunit of the ATP synthase, which recognizes oligomers of the enzyme, and provided values for the content of ATP synthase in the cell much larger than those of PLA signals revealing the interaction between β-F1-ATPase and IF1, supporting that the fraction of sluggish ATP synthase in cells is less than in tissues such as brain and kidney. The co-

distribution of β-F1-ATPase and IF1 immunoreactivity and the preferential localization of β-F1-ATPase at cristae tips, supports that the sluggish ATP synthase is preferentially located at the tips because tetrameric assemblies of IF1-inhibited ATP synthase are likely to stabilize the rows of ATP synthase at this location. However, we should take into consideration that in BN-gels IF1 co-migrates with monomeric and oligomeric assemblies of the

**Fig. 5 Intramitochondrial distribution of ATP synthase and IF1 in mouse tissues. a** Representative electron micrographs of mouse kidney and heart mitochondria immunolabeled for β-F1-ATPase (up) or IF1 (down). The histograms show the mean ± SEM of β-F1-ATPase ($n = 729–925$) and IF1 ($n = 153–225$) gold particles per mitochondrial area ($n = 33–40$ mitochondria). *$p < 0.05$ when compared to heart by Student's $t$ test. **b** Histograms show β-F1-ATPase abundance (gold particles/μm) in crista sheets or tips ($n = 27–38$). ATP synthase labeling of the sheet represents both enzyme at the sheet and at the rim of the sheet, because ATP synthase decorates cristae rims all along[45]. Hence, the labeling at the sheet is an overestimation of enzyme at this location. ***$p < 0.001$ when compared to sheet abundance by Student's $t$ test. **c** Representative electron micrographs of mouse kidney, brain, liver and heart mitochondria immunolabeled for β-F1-ATPase (10 nm gold) and IF1 (15 nm gold). Black arrowheads point to two or more β-F1 particles not further than 15 nm. Red arrowheads show IF1 and β-F1 particles not further than 15 nm. **d** Histograms show the number β-F1 (left axis, blue bars) or IF1 (right axis, orange bars) particles per mitochondrial area in mouse kidney, heart, brain and liver. $p < 0.0001$ when compared to kidney, heart or brain and #$p < 0.001$ when compared to kidney and brain by one-way ANOVA and Dunnett's multiple comparison tests. *$p < 0.01$ when compared to heart by Student's $t$ test. **e** Histograms showing the percentage of β-F1 particles in doublets or triplets in mouse kidney, heart, brain and liver mitochondria. **f** Histograms depict the fraction of IF1 particles close to β-F1 particles. **d–f** Range of IF1 ($n = 37–217$) and β-F1-ATPase ($n = 591–1978$) gold particles in mitochondria ($n = 54–69$) per tissue. The histograms show the mean and the error bars ± SEM. Tissues from two mice were processed. See also Supplementary Fig. 4.

ATP synthase suggesting that IF1 could bind and inhibit the enzyme indistinctly of its assembly. Results in CN-gels confirmed that as long as IF1 is bound to any of the presentations of ATP synthase that fraction of enzyme will be inhibited as revealed by the sharp increase of ATP hydrolysis in the IF1-depleted conditions.

We show that the interaction of IF1 with the ATP synthase defines in part the bioenergetic function of mitochondria by controlling the rate of respiration, ΔΨm and mtROS production. Indeed, partial blockade of ATP synthesis by IF1 restrains electron flow through respiratory complexes in coupled mitochondria and, consistently, its ablation releases the restriction in cellular rates of respiration. Interestingly, the increased flux of respiration does not involve changes in the expression of proteins of the respiratory chain but it is exerted by an enhanced assembly of the supercomplex, in agreement with the prominent role that this superassembly of the respiratory chain plays in favoring electron transfer to minimize mtROS production[50]. Since cristae length is smaller in IF1-KO cells, it is tempting to suggest that the reduced surface of cristae sheets could favor SC formation by increasing the density of respiratory complexes. Likewise, we show that the IF1-mediated blockade of ATP synthase in control cells contributes to mitochondrial hyperpolarization, because it prevents the backflow of protons into the matrix and contributes to mtROS production, which is function of ΔΨm values[42]. Both ΔΨm and ΔpH, the two components that build the proton motive force, are heterogeneously distributed throughout mitochondria[39,40,51]. Remarkably, and consistent with the overall IF1-mediated changes in ΔΨm and mtROS production observed, STED super-resolution microscopy of mitochondria indicated that the sites of higher IF1 content significantly co-distribute with the sites of higher ΔΨm, stablishing a close link between IF1 and the bioenergetic activity of the organelle. Interestingly, similar IF1-mediated changes in respiration, ΔΨm and mtROS production have been reported in neurons derived from mouse models of loss and gain of function of IF1[6], further supporting the in vivo role of IF1 as a relevant regulator of mitochondrial bioenergetics under basal conditions.

By in situ live cell microscopy, it has been suggested that mitochondria contain two subpopulations of Complex V, ATP synthase preferentially located at cristae edges and ATP hydrolase in the cristae sheet[51]. Obviously, this situation would result in futile high-energy dissipating cycle. The same lab has suggested that under steady-state OXPHOS the local proton motive force at sites of active ATP synthase is low and, IF1 is needed to block ATP hydrolysis by the enzyme[40]. Rather than supporting the co-existence of ATP synthase and ATP hydrolase in coupled mitochondria, and because of the structural and functional roles played by IF1 in oligomerization and inhibition of both ATP synthase activities, we support that under OXPHOS conditions

there are two pools of ATP synthase, active and inactive (sluggish). The heterogeneity of ΔΨm and perhaps of mtROS production resulting in part from IF1-mediated active/inactive state transitions of the sluggish ATP synthase. Changes in local pH, ΔΨm, $Ca^{2+}$ and other ions[3,10,15,32] and post-translational modifications, including phosphorylation of IF1[28,29] or of other ATP synthase subunits[52,53] are likely to mediate state transitions of the enzyme in a highly dynamic manner to explain the rapid changes in ΔΨm within cristae. Since IF1 is a regulatory subunit of ATP synthase with a very short half-life (~3 h)[18] when compared to other structural subunits of the enzyme (~18 h)[54], we support that regulation of IF1 turnover might also participate in the regulation of the sluggish ATP synthase under physiological conditions.

Importantly, IF1 is tissue-specifically expressed[20] and its effects on the physiological response of the tissues vary depending upon the tissue in which it is being expressed. For instance, the overexpression of IF1 in brain prevents excitotoxic damage to neurons[23] or improves cognition[6]. Likewise, the overexpression of IF1 in intestinal cells alleviates from inflammatory insults[25]. However, the expression IF1 in mouse tissues devoid of IF1 (skeletal muscle, liver) or with low IF1 content (heart) is detrimental[24,26]. In fact, the overexpression of IF1 in mouse heart by knocking out *Lrpprc* gene results in progressive lethal cardiomyopathy by bioenergetic impairment[55]. Mechanistically, LRPPRC interacts with IF1 mRNA[56] acting as a negative regulator of its translation both in mouse and human cells[20]. Likewise, IF1 is upregulated in cardiac hypertrophy induced by pressure overload, myocardial infarction, or α-adrenergic stimulation, showing the hypertrophied hearts increased nonproductive tetramers of inhibited ATP synthase[27]. Consistently, IF1-knockout mice are protected from pressure overload induced cardiac hypertrophy[57]. In the same line, the expression of IF1 in skeletal muscle of mice promotes an impaired bioenergetic function of mitochondria resulting in a phenotype prone to insulin resistance and metabolic syndrome at short term[26] and in a tubular aggregate myopathy in the long term[58]. Likewise, the expression of IF1 in mouse liver, that promotes oligomeric assemblies of ATP synthase[24], is pro-oncogenic facilitating tumor development[24] in agreement with findings in human hepatocarcinomas[59]. Overall, we support that mouse models of loss and gain of function of IF1 will certainly contribute to delineate the precise mechanisms underlying the tissue-specific functions of the inhibitor of the ATP synthase.

## Methods

**Mice**. Mice maintained on C57BL/6 J background, were housed at the CBMSO Animal Facility with a 12-h light/12-h dark cycle at 18–23 °C with 40–60% humidity. Global IF1 knockout mice were obtained by breeding IF1-floxed mice[6] with B6.Cg-Tg(Camk2a-cre)T29-1Stl/J mice (The Jackson Laboratory, JAX:005359). The latter

express the Cre recombinase in forebrain neurons and in the male germline[60]. Germline recombination of the floxed IF1 allele was assessed by PCR in different tissues and its inheritance by breeding with wild-type C57BL/6 J mice (The Jackson Laboratory, JAX:000664). Experiments were carried out with 10-week-old mice.

The genotyping strategy for genetically modified IF1 mice has been described[6]. Wild-type and IF1-floxed alleles were distinguished with 5'-TGCCTGACATTGG TATTGGG-3' and 5'-GTGCAGCTTGTGGGAGTCAG-3' primers. The transgene encoding the Cre recombinase was detected with 5'-CATTTGGGCCAGCTAA ACAT-3' and 5'-TAAGCAATCCCCAGAAATGC-3' primers. The recombined IF1-KO allele was genotyped with the primers 5'-AAGGCGCATAACGATAC CAC-3' and 5'-ACTGATGGCGAGCTCAGACC-3'.

**Ethics statement**. Mouse experiments were carried out in accordance with EU Directive 2010/63/EU for animal experiments. The Ethics Committee of Animal Experimentation (CSIC-UAM, CM PROEX 233/19) and The Institutional Review Board of UAM (CEI 101-1891-A325) approved the project.

**Cell lines and culture conditions**. Human colorectal carcinoma HCT116 cell lines (ATCC CCL-247) were grown in McCoy's 5 A medium supplemented with 10% fetal bovine serum (FBS) and cultured in an incubator at 37 °C with ambient air 10% $CO_2$. Human T cell leukemia Jurkat cell lines (ATCC TIB-152) were grown in RPMI supplemented with 10% FBS. Normal rat kidney (NRK) cells (ATCC CRL-6509) were grown in DMEM supplemented with 10% FBS as indicated above. Rates of cellular proliferation were monitored by counting the cells every day during three days after seeding 200,000 cells per well in a 6-well plate.

**Cloning strategies**. Primers for guide RNAs targeting human *ATP5IF1* (sg1RNA-IF1, 5'-CACCGAATGGCAGTGACGGCGTTGG-3' and 5'-AAACCCAACGCCG TCACTGCCATTC-3', or sg2RNA-IF1, 5'-CACCGCGGACGTGGCTTGGCGT GTG-3' and 5'-AAACCACACGCCAAGCCACGTCCGC-3') were annealed and cloned into pSpCas9(BB)-2A-GFP (PX458, Addgene, 48138) plasmid using the *Bbs*I sites. A non-targeting guide RNA sequence (sgRNA-scramble, 5'-CACCGG-TAGCGAACGTGTCCGGCGT-3' and 5'-AAACACGCCGGA-CACGTTCGCTACC-3') was used as a control.

To generate the fusion protein IF1-GFP, the coding sequence of the human *ATP5IF1* was amplified by PCR (Expand High Fidelity Plus PCR System, Roche) using pCMV-SPORT6-IF1 plasmid[16] as a template. The IF1-GFP primers (5'-CTACCGGACTCAGATCTCGAGATGGCAGTGACGGCGTTG-3' and 5'-CCGGTGGATCCGGGCCCTATCATCATGTTTTAGCATTTTG-3') add *Xho*I and *Apa*I restriction sites in forward and reverse, respectively. The resulting product was purified and first cloned into pGEM-T easy vector (Promega, A1360) and after subcloned into pEGFP_C3 (Clontech, 6082-1) in the *Xho*I and *Apa*I restriction sites. The resulting plasmid, pIF1-GFP, encodes a fusion protein consisting of human IF1 with a C-terminal fusion of GFP and was verified by Sanger sequencing. The phosphomimetic mutant of IF1, unable to bind ATP synthase[28], was generated by site-directed mutagenesis with QuickChange Lightning Kit (Agilent Technologies) using the pIF1-GFP plasmid as template and primers S39E (5'-CCGGGGCGCGGGCGAGATCCGGGAAGCCG-3' and 5'-CGGCTTCCCGGATCTCGCCCGCGCCCCGG-3'). The resulting mutant IF1-S39E-GFP was verified by Sanger sequencing.

**Cell transfection**. The generation of stable IF1-KO HCT116 or Jurkat cells was carried out by the CRISPR-Cas9 technique[61]. In brief, cells were transfected with the plasmids expressing scramble- or IF1-targeting guide RNAs using Lipofecta-mine 2000 reagent (Invitrogen, 11668019) and sorted by GFP expression (FAC-SAria Fusion, at Flow Cytometry Service from CBMSO) 48 h after transfection. Sorted cells were expanded and the absence of IF1 expression was confirmed by Western Blot and immunofluorescence.

HCT116 cells were transfected with plasmids expressing EGFP targeted to mitochondrial by N-terminal fusion with β-F1-ATPase targeting sequence (pβ-GFP-3'UTR)[41], IF1-GFP or IF1-S39E-GFP fusion proteins using Lipofectamine 2000 reagent (Invitrogen, 11668019). Cells stably expressing pre-β-GFP were selected using 40 μg mL$^{-1}$ of geneticin (Gibco, 10131035), and cells expressing IF1-GFP or IF1-S39E-GFP with 100 μg mL$^{-1}$ bleomycin (Sigma-Aldrich, B7216). Expression of the constructs was assessed by live cell imaging, Western Blot and immunofluorescence.

NRK cells were transfected using Lipofectamine 2000 reagent (Invitrogen, 11668019) with pCMV-SPORT6-IF1, the phosphodeficient (S39A) or phosphomimetic (S39E) IF1 mutants[28]. Cells were harvested 24 h post-transfection and processed for 2D-gel electrophoresis.

**Antibody production**. Antibodies against recombinant human IF1 were produced as described for the mouse counterpart[20]. In brief, New Zealand White rabbits (Charles River, Crl:KBL) were immunized with 4 doses of the purified protein (200 μg dose$^{-1}$) emulsified (1:1) with complete (first dose) or incomplete (3 final doses) Freund adjuvant (Sigma-Aldrich) every two weeks. Serum immunoreactivity was tested by Western Blot using recombinant human IF1 and human cellular extracts. The specificity of the antibody was validated in IF1-KO cells generated in this paper.

**Isolation of mitochondria and cytosol**. Mitochondrial isolation was performed as described previously[28]. HCT116 or Jurkat cells were homogenized in a glass-Teflon homogenizer in seven volumes of hypotonic buffer (83 mM sucrose, 10 mM MOPS pH 7.2). Following homogenization, the same volume of hypertonic buffer (250 mM sucrose, 30 mM MOPS pH 7.2) was added. Tissues were homogenized in cold buffer A (320 mM sucrose, 1 mM EDTA, 10 mM Tris-HCl pH 7.4). Mito-chondria were pelleted by centrifugation at $7500 \times g$. The supernatants were additionally centrifuged at $50,000 \times g$ for 30 min at 4 °C to obtain the cytosol.

**Determination of ATP synthetic activity of ATP synthase**. Digitonin-permeabilized HCT116 or Jurkat cells ($2 \times 10^6$ cells) or freshly isolated mouse mitochondria (20–50 μg) were used for determining the oligomycin-sensitive mitochondrial production of ATP[62,63]. Cells were permeabilized with 50 μg mL$^{-1}$ digitonin in respiration buffer (225 mM sucrose, 10 mM KCl, 5 mM MgCl$_2$, 0.05% w/v BSA, 10 mM potassium-phosphate buffer, 1 mM EGTA and 10 mM Tris-HCl; pH 7.4) supplemented with EDTA-free protease and phosphatase inhibitory cocktails (MilliporeSigma). Permeabilized cells, or mouse mitochondria previously resuspended in respiration buffer without digitonin, were directly added to the luminometer plate reader (Omega FLUOstar, BMG LABTECH). ATP production in the absence or presence of 2 μM oligomycin was measured as luminescence production in respiration buffer containing 0.1 mM ADP, 5 mM succinate, 0.15 μM P1,P5-di(adenosine-5′) pentaphosphate, 0.165 mg mL$^{-1}$ of luciferin and 0.003 mg mL$^{-1}$ luciferase. The luminometer plate-reader was set up to measure luminescence in kinetic mode every 15 sec during ~8 min. Luminescence is the product of the luciferin converted to oxyluciferin in the presence of ATP and increases linearly with the production of ATP (Fig. 1c). Relative light units are converted to ATP concentration using an ATP standard curve prepared in respiration buffer (Fig. 1c)[62,63]. The ATP produced was calculated from the slopes in the absence of oligomycin after subtracting the slopes in the presence of oligomycin.

**Determination of ATP hydrolytic activity of ATP synthase**. Isolated mito-chondria from HCT116, Jurkat cells or mouse tissues were used for the spec-trophotometric determination of the hydrolytic activity of ATP synthase[63,64]. Determination of ATP hydrolytic activity is assayed in isolated broken mito-chondria by three cycles of freezing and thawing the preparations in liquid nitrogen and at 37 °C, respectively. 20 μg of isolated mitochondria were resus-pended in reaction buffer (50 mM Tris-HCl, pH 8.0; 5 mg mL$^{-1}$ BSA, 20 mM MgCl$_2$, 50 mM KCl) supplemented with EDTA-free protease and phosphatase inhibitory cocktails (MilliporeSigma). ATP hydrolysis was determined in kinetic mode in reaction buffer containing 5 μM FCCP, 1 μM antimycin A, 10 μM PEP, 2.5 mM ATP, 1 mM NADH, 4 units of LDH and 4 units of PK in a luminometer (Omega FLUOstar, BMG LABTECH) in a final volume of 100 μl per well. In this reaction, the ADP generated by the hydrolysis of ATP allows PK to transform phosphoenolpyruvate (PEP) into pyruvate, at a ratio of 1 mole of ADP:1 mole of PEP:1 mole of pyruvate. Pyruvate is then used by LDH to generate lactate oxidizing a molecule of NADH to NAD$^+$. By each ATP molecule hydrolyzed one molecule of NADH is oxidized. The specificity of the assay was assessed by the addition of 2 μM oligomycin, that was added to some wells at the beginning of the reaction or within few minutes of starting the reaction but before exhaustion of the reagents. The ATP hydrolase activity is calculated from the slopes of the kinetics after subtracting the slopes obtained in the presence of oligomycin and converting ΔA$_{340}$ nm/min into nmoles of ATP using the molar extinction coefficient of NADH ($\varepsilon_{340nm} = 6.22 \times 10^{-3}$ M$^{-1}$ cm$^{-1}$).

**Mitochondrial enzyme activities**. Isolated mitochondria from HCT116 cells were used for the spectrophotometric determination of the activity of complexes I, II and IV[64], after three cycles of freezing and thawing in liquid nitrogen and at 37 °C, respectively. Complex I activity (NADH:ubiquinone oxidoreductase) was measured at A$_{340}$ using 100 μg of mitochondria in 1 mL C1/C2 buffer (25 mM K$_2$HPO$_4$, 5 mM MgCl$_2$, 3 mM KCN and 2.5 mg mL$^{-1}$ BSA) containing 0.1 M NADH (elec-tron donnor), 0.1 mM UQ1 (electron acceptor) and 1 mg mL$^{-1}$ antimycin A (Complex III inhibitor to prevent UQ1 re-oxidation). The complex I enzyme-catalyzed oxidation of NADH is monitored by following the decrease in the absorbance of NADH at 340 nm. Inhibition of the activity was accomplished by the addition of 1 μM rotenone. Complex II activity was measured at A$_{600}$ using 100 μg of mitochondria in 1 mL C1/C2 buffer containing 30 μM 2,6-dichlor-ophenolindophenol (DCPIP), 1 μM rotenone (Complex I inhibitor), 1 μM anti-mycin A (Complex III inhibitor), 10 mM succinate (Complex II substrate) and 6 mM phenazine methosulfate (intermediate electron acceptor). The complex II enzyme-catalyzed reduction of DCPIP is measured by following the decrease in absorbance at 600 nm due to the oxidation of DCPIP[65]. Complex IV activity was determined at A$_{550}$ using 100 μg of mitochondrial protein in 10 mM KH$_2$PO$_4$, pH 6.5, 0.25 M sucrose, and 1 mg mL$^{-1}$ BSA containing 10 μM reduced cytochrome c as electron donor. Complex IV activity is measured by following the decrease in absorbance at 550 nm due to the oxidation of cytochrome c. Addition of 240 μM KCN was used to inhibit enzyme activity.

**Analysis of cellular respiration and rates of glycolysis**. Oxygen consumption rates (OCRs) were determined in Seahorse XF24 and XFe96 Extracellular Flux Analyzers (Agilent Technologies). HCT116 or Jurkat seeded cells (30,000 cells/well) were equilibrated with Seahorse XF base medium (Agilent Technologies) supplemented with 10 mM glucose, 2 mM glutamine and 1 mM pyruvate for 1 h before assay at 37 °C in a $CO_2$-free incubator. Mitochondrial function was determined through sequential addition of 6 μM oligomycin, 1 μM 2,4-dinitrophenol (DNP) and 1 μM antimycin A plus 1 μM rotenone following the XF Cell Mito Stress Test injection protocol designed by the manufacturers.

To determine the rates of glycolysis, the initial rates of lactate production by the cells were determined by the enzymatic quantification of lactate concentrations in the culture medium[16]. Culture medium was replaced by fresh medium supplemented with 1% FBS. Samples of culture medium were taken at different intervals (0, 2, 4 and 6 h) and precipitated with 4 volumes of cold perchloric acid, incubated on ice for 1 h and then centrifuged for 5 min, $11,000 \times g$ at 4 °C to obtain a protein-free supernatant. The supernatants were neutralized with 20% (w/v) KOH and centrifuged at $11,000 \times g$. Lactate levels were spectrophotometrically determined (Omega FLUOstar, BMG LABTECH) following the reduction of $NAD^+$ at $A_{340}$ after the addition of 4 units of LDH (MilliporeSigma, 10127230001).

**Determination of ΔΨm and mtROS production**. ΔΨm and mtROS production were determined in HCT116 cells by flow cytometry after staining the cells with 50 nM TMRM (Invitrogen, T668) or 2.5 μM MitoSOX (Invitrogen, M36008) probes, respectively[17]. The fluorescence intensity of at least 10,000 events in DAPI unstained cells was determined in a FACS Canto II cytometer (Becton Dickinson). Specificity of TMRM staining was assessed by the addition of 1 μM FCCP once recorded the basal TMRM fluorescence. For ATP synthase inhibition, 2 μM oligomycin was added during TMRM or MitoSOX staining. For inhibition of multidrug resistance channel, 1.6 μM cyclosporin H was added in the culture medium of CRL and IF1-KO cells 24 hours before assessing ΔΨm. The analysis was performed using the FlowJo software (Tree Star). The gating strategy consisted in selecting cells by its size and complexity (Supplementary Fig. 5a1), exclusion of doublets (Supplementary Fig. 5a2), live-cells selection by exclusion of DAPI-staining (Supplementary Fig. 5a3) and the mean fluorescence intensity of TMRM represented (Supplementary Fig. 5a4). The same procedure was followed for mtROS production using MitoSOX staining (Supplementary Fig. 5b1-4).

**Western blot analysis**. Cells, isolated mitochondria or mouse tissues were resuspended in Tissue Protein Extraction Reagent (T-PER, Thermo Fisher, 78510) supplemented with EDTA-free protease and phosphatase inhibitory cocktails (MilliporeSigma). Homogenates were freeze-thawed three times in liquid nitrogen and clarified by centrifugation at $11,000 \times g$ for 30 min at 4 °C. Protein concentration was determined using Bradford reagent (Bio-Rad protein assay, Bio-Rad). The resulting protein extracts were fractionated on SDS-12% PAGE and transferred onto PVDF (Merck KGaA, IPVH00010) or nitrocellulose (GE Healthcare, 15289804) membranes for immunoblot analysis. Membranes were blocked with 5% nonfat dried milk in TBS with 1% Tween 20 for 1 h and incubated with the primary antibodies diluted in 3% BSA in TBS overnight at 4 °C. The primary antibodies used are listed in the Supplementary Table 1. As secondary antibodies, anti-mouse or anti-rabbit IgGs (1/5000) (Nordic Immunology) were used diluted in TBS with 1% Tween 20. Precision Plus Protein Standards Kaleidoscope (Bio-Rad) were run in the gels and used to infer the band size of the proteins investigated. The use of monospecific antibodies also helped in band identification. Protein bands were visualized using the Novex® ECL (Thermo Fisher). The intensity of the bands was quantified using a GS-900™ Calibrated Densitometer (Bio-Rad) and the Analyze Gel command of ImageJ software (NIH).

**2D, Blue-Native (BN) and Clear-Native (CN) gel electrophoresis**. Isoelectrofocusing (IEF) was performed with 13-cm Immobiline DryStrips of 6–11 L [linear] pH gradient using an Ettan IPGPhor3 IEF unit (GE Healthcare)[28]. In brief, 200 μg of cellular protein diluted in 250 μL of DeStreak Rehydration Solution (GE Healthcare, 17600319) containing 0.5% of the corresponding IPG buffer (GE Healthcare, 17600178) were loaded in the 13 cm strips. Strips were rehydrated at 20 °C during 12 h at 50 V, and then, focused 15 min at 250 V, 1 h at 500 V, 1 h at 1000 V, followed by a lineal increase during 3 h until 10,000 V which were maintained to achieve 60000 V-h. After the IEF strips were equilibrated with 0.170 M Tris-HCl pH 6.8 containing 6 M urea, 20% glycerol, 2% SDS and 130 M DTT for 15 min, followed by another 15 min incubation in the same buffer changing the DTT by 135 mM iodoacetamide with Bromophenol Blue. The equilibrated strips were transferred to the top of an SDS-12% PAGE gel. Electrophoresis was carried out using a Protean II XI system (Bio-Rad) with constant current (30 mA) at 4 °C for 3 h. Western Blot analysis of the fractionated proteins was performed as described previously.

For BN and CN gels, mitochondrial pellets from HCT116 cells and mouse tissues were processed in 50 mM Tris-HCl pH 7.0 containing 1 M 6-aminohexanoic acid at a final concentration of 10 mg protein mL$^{-1}$. Membranes were solubilized by the addition of 10% digitonin using a ratio 4:1 digitonin:mitochondrial protein. 5% Serva Blue G dye in 1 M 6-aminohexanoic acid was added to the solubilized membranes for BN gels. Serva Blue G dye was replaced by 0.1% Ponceau Red and 5.5% glycerol for

CN gels. Precast Native PAGE™ Novex® 3–12% Bis-Tris Protein Gels (Life Technologies, BN1001BOX) were used for BN and CN gels. Both BN and CN gels, were run at 70 V for 15 min, followed by 1 h at 10 mA. BN cathode buffer: 50 mM Tricine, 15 mM Bis-Tris, pH 7.0, 0.02% Serva blue G; CN-cathode buffer: 50 mM Tricine, 15 mM Bis-Tris, 0.05% sodium deoxycholate, pH 7.0; BN and CN-anode buffer: 50 mM Bis-Tris, pH 7.0. After fractionation, CN gels were incubated with 270 mM glycine, 35 mM Tris, 8 mM ATP, 14 mM $MgSO_4$, 0.2% $Pb(NO_3)_2$, pH 8.4 to assess the hydrolytic activity of the ATP synthase, which correlates with the formation of white precipitates of lead phosphate as a result of ATP hydrolysis[66]. After the fractionation of BN gels, the gels were electroblotted onto PVDF membranes and further processed for immunoblotting. The primary antibodies are listed in the Supplementary Table 1.

**Immunocapture of ATP synthase**. ATP synthase was immunopurified from isolated mitochondria of HCT116 cells or mouse tissues using the commercial ATP Synthase Immunocapture Kit (Abcam, ab109715) according to the manufacturer's instructions. Briefly, 1.5–3 mg of mitochondria were solubilized with 1% n-dode-cyl-β-D-maltoside, centrifuged to remove insoluble material and incubated with the affinity beads at 4 °C overnight. The affinity beads were washed twice before ATP synthase was eluted and fractionated on SDS-12% PAGE. After fractionation, the gels were electroblotted onto nitrocellulose membranes and further processed for immunoblotting. β-F1-ATPase was detected with rabbit anti-β-F1-ATPase antibody[44] labeled with Cy5 using Amersham CyDye Reactive Dye Pack (GE Healthcare, PA25001) to visualize the protein avoiding the detection of the heavy chain of the immunoglobulins. IF1 was detected with rabbit anti-human (this study) or anti-mouse IF1 antibodies. GFP was detected with rabbit anti-GFP antibody (see Supplementary Table 1).

**Immunofluorescence and Proximity Ligation Assay (PLA) experiments**. For immunofluorescence, cells cultured on coverslips were fixed with 4% paraformaldehyde (PFA) and then permeabilized with 0.1% Triton X-100. Cells were blocked with 1% BSA, 0.1% TritonX-100 in PBS and incubated with the corresponding primary antibodies (see Supplementary Table 1) at 4 °C overnight. Appropriate secondary antibodies were used (see Supplementary Table 1). Samples were mounted in Mowiol with DAPI.

For Proximity Ligation Assays (PLA), Duolink® PLA Probes and Fluorescent Detection Reagents (Sigma-Aldrich, DUO92004, DUO92002, DUO92014) were used following manufacturer's protocol. In brief, fixed and permeabilized HCT116 and Jurkat cells were blocked with Duolink Blocking Solution. For assessing the interaction between β-F1-ATPase and IF1, cells were incubated with primary mouse anti-β-F1-ATPase (1/500) and rabbit anti-IF1 (1/1000) antibodies (see Supplementary Table 1). After overnight (o/n) incubation, the PLA probe antibodies rabbit-PLUS and mouse-MINUS were added for 1 h at 37 °C. For assessing supercomplex formation, cells were incubated with primary rabbit anti-NDUFS5 (1/500) and mouse anti-UQCRC2 (1/500) antibodies (see Supplementary Table 1). After o/n incubation, the same PLA probe antibodies were used. For ATP synthase oligomerization, samples were incubated o/n with anti-γ-F1-ATPase antibody (1/250)[67] as unique primary antibody. After o/n incubation, PLA anti-mouse PLUS and anti-mouse MINUS probes were added as secondary antibodies. For amplification, the samples were processed following manufacturer's instructions. Samples were mounted with in situ DAPI-containing mounting medium. Omission of primary antibodies was used for assessing the specificity of the assays. For localization of PLA dots within the mitochondrial reticulum, cells were stained with 500 nM MitoTracker™ Red CM-H 2 Xros during 30 minutes before cell fixation. However, this procedure was omitted when PLA quantifications were done because mitochondrial staining significantly affected amplifications of PLA signals.

Confocal microscopy was carried out in an A1R+ microscope (Nikon) at CBMSO and analyzed with ImageJ software. Pearson's correlation coefficients were obtained with the JACoP plugin of ImageJ software. For the analysis of the PLA signals per cell a self-designed macro was used. In brief, the maximum intensity of the z stacks comprising the cells was projected and the number of dots and nuclei were counted with the command analyze particles in thresholded images.

**High resolution live cell imaging**. HCT116 cells were seeded in IBIDI μ-Dish 35 mm HIGH Glass Bottom dishes and stained during 30 min with 50 nM TMRM (Tetramethylrhodamine Methyl Ester; Invitrogen, T668) or 2.5 μM MitoSOX (Invitrogen, M36008). To avoid phototoxicity, we eliminated phenol red and FBS from the incubation media by incubating cells with HBSS medium supplemented with 10 mM glucose prior to live-cell imaging. Cells were maintained at fixed $CO_2$ and temperature and the fluorophores excited with the less possible power of the laser.

For high resolution live imaging acquisition, a Leica SP8 LSM, fitted with STED module at CNB Advanced Light Microscopy Facility was used with an immersion objective HCX PL APO CS 100x NA 1.4. Pixel size and z-stacks step distances were determined with the Nyquist calculator (Scientific Volume Imaging) to obtain optimal conditions for deconvolution. For GFP acquisition, a HyD 2 detector was used (488/498-528 nm wavelengths for excitation/collection with a white laser) applying a 592 nm STED depletion laser. For TMRM acquisition, a HyD 5 detector

was used (552/564-668 nm wavelengths for excitation/collection with a white laser) applying a 660 nm STED depletion laser. For MitoSOX acquisition, a HyD 5 detector was used (552/560-632 nm wavelengths for excitation/collection with a white laser) applying a 660 nm STED depletion laser. Before image analysis, raw.lif files were automatically corrected for its chromatic aberration with the full correction-iterative fit algorithmic and deconvoluted using Huygens deconvolution software (Scientific Volume Imaging). Deconvoluted images were then analyzed using the JACoP plugin of ImageJ (NIH) to calculate the Manders' colocalization coefficients comparing the staining of GFP and TMRM pixel by pixel in entire images that contained on average 17 mitochondria. The only parameter of the JACoP plugin that we selected for the analysis was the threshold of the intensity of the staining for each fluorophore. In both cases, we choose a restrictive threshold selecting only the maximal fluorescence pixels of GFP and TMRM, which was the same for all the images. The M1 and M2 Manders' colocalization coefficients are provided by the JACoP plugin.

**Electron microscopy of HCT116 cells**. Cells were fixed in culture plates with 4% PFA and 2% glutaraldehyde in 0.1 M phosphate buffer (pH 7.4) during 2 hours at room temperature at the CBMSO Electron Microscopy Facility. Samples were postfixed in 1% $OsO_4$ plus 1% $K_3Fe(CN)_6$ in water, dehydrated with ethanol and after embedded in Taab 812 epoxy resin (TAAB Laboratories, E202). Resin polymerized sheets of cell monolayers were detached and fitted onto resin blocks to obtain orthogonal 80 nm ultrathin sections. Sections were transferred into slot grids and stained with uranyl acetate and lead citrate. A Jeol JEM1400 Flash Transmission Electron Microscope at 80Kv and a CMOS Oneview (4Kx4K) camera (Gatan) were used to obtain images. ImageJ (NIH) descriptors of mitochondrial shape and cristae were calculated. Each individual mitochondrion was surrounded using the freehand selection tool by following the outer membrane; the Area and Circularity were measured using the corresponding plugins under Measure command. Within each mitochondrion, cristae were counted using Multi-point tool and the length and width of each crista was calculated by drawing a line following the crista (from base to tip) or between both sides near the tip, respectively, and using the Measure command. Full cristae, in which there was no interruption between the base next to the outer membrane and the tip, were considered. Data shown refer to the average for all mitochondria or cristae.

**High resolution immunoelectron microscopy of mouse tissues**. Wild type C57BL/6 J mice (The Jackson Laboratory, JAX:000664) were transcardially perfused with 4% PFA in 0.1 M phosphate buffer, pH 7.4 (PB), and the tissues dissected and left immersed in 8% PFA in PB overnight at 4 °C. Tissue blocks were washed in PB and sectioned in a vibratome (Leica, VT1200S) to obtain 300 μm width slices. Tissue sections were incubated twice with 0.05 M $NH_4Cl$ for 20 min each at room temperature to quench free aldehydes, cryoprotected in an increasing series of glycerol concentrations (10, 20, 30% glycerol in PB) and stored in 30% glycerol at 4 ˚C. Freeze substitution was performed essentially as described[68]. In brief, sections were plunge frozen in liquid ethane in a Reichert Jung KF80 unit (Leica) and then immersed in 0.5% uranyl acetate in methanol at –90 °C in a Leica AFS freeze-substitution device. The freeze-substitution conditions were as follows: –90 °C for 80 h, temperature increased up to –50 °C by 4 °C/h for 10 h and then 83 h at –50 °C. Sections were washed with methanol to eliminate uranyl acetate and infiltrated with Lowicryl HM 20 resin (Polysciences, 15924-1) at –50 °C. The resin was polymerized by indirect long-wavelength UV-irradiation (360 nm) at –40 °C for 38 h and at 22 °C for 24 h.

Ultrathin sections (70 nm) were obtained using an Ultracut E ultramicrotome (Leica) and subjected to post-embedding gold immunolabeling. Sections were first incubated with 0.15% glycine in PBS for 15 min to quench free aldehyde groups, followed by blocking buffer (0.1% BSA, 1% cold water fish skin gelatin in Tris-buffered saline (30 mM Tris, 150 mM NaCl, pH 8.2)) for 30 min at room temperature. For single immunolabeling, sections were incubated for 1 h with primary antibodies (rabbit anti-IF1[20] 1/5 or rabbit anti-β-F1-ATPase[44] 1/25) diluted in 0.1% BSA-c (Aurion, 900.099) in Tris-buffered saline. Then, sections were washed 3 times with the antibody diluent solution, incubated for 45 min with PAG 10 nm (Protein A Gold, PAG 10 nm/S), washed twice with Tris-buffered saline and 3 times with ddH$_2$O. Staining was carried out with uranyl acetate and lead citrate.

Double immunolabeling was performed sequentially[44]. Tissue sections were first incubated with rabbit anti-IF1[20] 1/5 and then with PAG 15 nm (PAG 15 nm/S) as described above. They were washed 3 times with antibody diluent solution (30 min each wash) and incubated with 0.1 mg mL$^{−1}$ Protein A in Tris-buffered saline for 30 min to block possible exposed binding sites from the first antibody. Then, sections were washed 3 times in Tris-buffered saline, blocked again, incubated with rabbit anti-β-F1-ATPase[44] 1/25, washed, incubated with PAG 10 nm (PAG 10 nm/S), washed, and stained as described for the single immunolabeling.

All the preparations were examined at 80Kv in JEM-1400Flash transmission electron microscope (JEOL) coupled to a CMOS digital camera OneView (4Kx4K) (Gatan). Samples were processed and imaged at the CBMSO Electron Microscopy Facility. Controls included absence of primary antibody and showed little or no gold labeling. Moreover, in the double immunolabeling, a control omitting the second antibody (anti-β-F1-ATPase[44]) was included to confirm that PAG 10 nm

did not bind to the first antibody (anti-IF1[20]). Background gold labeling was assessed by counting particles in the nuclei and subtracted to the measurements of gold particle density in mitochondria.

Images were analyzed in ImageJ. Particles were localized in each mitochondrial subcompartment (inner boundary membrane, cristae junctions, cristae sheets or tips) by visually inspecting which membrane they were close to. As a cutoff, we considered a distance lower than 15 nm, which accounts for the size of the antibody, which is about 10 nm, and approximately half of the gold particle. Particles at a longer distance or in blurry areas were not considered. The total length of the membranes corresponding to cristae tips or sheets for each mitochondrion was calculated by drawing a line on each one and using the Measure command. The density of particles in cristae tips and sheets was calculated by dividing the number of cristae in each subcompartment by the corresponding length of membrane. Particle density was calculated for each mitochondrion by dividing the total number particles by the mitochondrial area, which was calculated as described above. When analyzing IF1 particles close to β-F1-ATPase particles or doublets of the latter, only particles closer than 15 nm to each other were considered.

**Statistics and reproducibility**. Each experiment was performed at least three independent times with the same results except the determination of complex II activity which has been performed only twice. Statistical analyses were performed by two-sided Student's t-test or one-way ANOVA with a post hoc Dunnett test, calculated using GraphPad Prism v8, and the details used in each experiment can be found in the figure legends. $P < 0.05$ values were considered statistically significant. Statistical $P$ values are provided in figure legends ($*P < 0.05$, $**P < 0.01$, $***P < 0.001$). The n used in each statistical test is indicated in the Figure Legends. When not specified, n refers to the animals or sample size per genotype.

**Reporting summary**. Further information on research design is available in the Nature Portfolio Reporting Summary linked to this article.

## Data availability

The authors declare that the data supporting the findings of this study are available within the paper and its Supplementary Information files that also contains Supplementary Figs. 6–11 with the uncropped blots of Figs. 1–4 and Supplementary Fig. 2. Should any raw data files be needed in another format they are available from the corresponding author upon reasonable request. Source Data for the figures are provided in Supplementary Data 1. A Reporting Summary for this article is available as a Supplementary Information file.

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

## Acknowledgements
We thank Beatriz Cicuéndez (CBMSO) and Brenda Sanchez (CBMSO) for invaluable help in the development of polyclonal IF1-antibody and PLA, respectively. María Teresa Villalba and Dr. Maite Rejas are acknowledged for their kind assistance in Confocal and Electron Microscopy studies at the Facilities of the CBMSO, respectively. The work was supported by grants from MINECO (PID2019-108674RB-100), CIBERER-ISCIII (CB06/07/0017) and Fundación Ramón Areces, Spain. IRC and PBEM were supported by predoctoral fellowships from FPU-MINECO and La Caixa, respectively. SDZ and CNT were supported by predoctoral fellowships from FPI-MINECO, Fondo Social Europeo.

## Author contributions
I.R.-C., P.B.E.-M., S.D..-Z. and C.N.-T., performed experiments and data collection; J.M.C. conceived the study, secured funding and wrote the paper. All the authors read, contributed, and approved the final manuscript.

## Competing interests
The authors declare no competing interests.
