## [Peer Review File · Communications Biology]

Reviewers' comments:

Reviewer #1 (Remarks to the Author):

The main conclusions of this manuscript are the existence of two ATP synthase pools that depend on IF1 binding, and the preferential location of IF1-dependent "sluggish" ATP synthase at the cristae tips.

As known from previous literature, IF1 is the canonical inhibitor of ATP synthase that limits ATP depletion under conditions of severe oxygen deprivation or the in presence of a dysfunctional respiratory chain, a specific function that is mainly modulated by acidic pH. However, new roles of IF1 in normoxic conditions are emerging to suggest that IF1 could also prevent ATP synthesis, although contrasting data have been reported so far. The role of IF1 in the oligomerization of ATP synthase is also controversial. The present work inserts in this scenario reporting that IF1 knock-out cells (cancer cell lines) show an increased rate of both ATP hydrolysis and synthesis, an increased basal respiration (which is explained by a higher amount of supercomplexes) and a reduction of ATP synthase oligomers. Authors also report that IF1 expression in different tissues correlates with the hydrolytic capacity of ATP synthase. Overall the text is fluent, the workflow is linear and data are clearly presented. Few typos are present (for example page 4, line 66 "affect" lacks the final "s"). I believe that the novelty of this work and the most interesting aspect is the hypothesis that IF1 promotes oligomerization of inactive ATP synthases within specific compartments (i.e. cristae tips) which correspond to hyperpolarized microdomains that would explain the heterogeneity of mitochondrial membrane potential. Nevertheless, I found that data provided in this manuscript do not entirely support this hypothesis, and I pointed out some issues that Authors should address before this work can be reconsidered for publication in Communications Biology.

Major points:

1. Figure 1, panel D, are the two cell lines equally sensitive to oligomycin? There is only one trace relative to the effect of oligomycin on ATP production and it is not clear to which cell line it refers.
2. Figure 1, panel E, it would be better to show the trace of mitochondria treated with oligomycin from the beginning (or a control trace with no oligomycin added) because as presented the drop of ATP hydrolysis could be due to exhaustion of the regenerating system. Are data of ATPase activity normalized to values obtained in presence of oligomycin?
3. Figure 2, panel A, I would suggest to provide values of basal and oligo-sensitive respiration that have been previously subtracted for the non-mitochondrial respiration, which clearly differs between the two genotypes. Why ablation of IF1 decreases membrane potential (panel C) and ROS production (panel D) needs to be addressed. If this is due to increased ATP production the difference should be abolished by the addition of oligomycin, and this needs to be tested. Further, the Authors should include cyclosporin H to inhibit MDR pumps to make sure that the TMRM loading is the same in the WT and KO cells.
4. The Authors state that the activity of respiratory chain complexes is unaltered between the two genotypes and thus the difference in basal respiration might be due to specific rearrangement of supercomplexes (SC). In Figure 2, panel E, SC are increased in membranes blotted against Complex I and IV components, while for CIII the difference is not evident at all (the spot on the top appears more an aspecific signal, could Author provide a better image?). I would also suggest to add loading controls for all presented BN gels. It would be important to provide a quantification of SC present. In the same panel, it seems that IF1 comigrates mostly if not entirely with ATP synthase monomers, although following Authors' hypothesis it should bind to and stabilize oligomers. Could Authors discuss this aspect?
5. In Figure 2, panel G, electron microscopy images show major alterations of IF1 KO mitochondria

which appear smaller and with fewer cristae than control (which is probably the most evident feature). Could Authors measure the number of cristae per mitochondria in the two genotypes? How can this be reconciled with an increased amount of SC?

6. Figure 3, Authors investigated whether presence of IF1 (and its binding to ATP synthase) correlates with a hyperpolarization by expressing a fusion IF1-GFP protein and a GFP (with the pre-sequence of subunit β) as control. Why is the GFP signal in pre β GFP-expressing cells almost the opposite to that of TMRM (panel F)? This would imply that GFP locates in depolarized compartments, is this the case? Then, it would be better to use a mutant of IF1 (fused to GFP) that does not bind ATP synthase as control for this experiment because it would normalize the effect of IF1 overexpression (likely present with IF1-GFP construct). I am a bit concerned by the fact that the two cell lines have been selected with different antibiotics (geneticin versus bleomycin) and therefore they are not really comparable. Moreover, Authors should check whether the fusion protein (IF1-GFP) can still interact with ATP synthase, which is not obvious. This is an important piece of information that needs to be provided. In general, I think that these findings do not demonstrate beyond reasonable doubt that IF1 localizes in hyperpolarized domains.

7. Figure 4, according to the Authors' hypothesis (IF1 binding to ATP synthase oligomers dampens enzymatic activity), I would expect that oligomers in BN gels should not show hydrolytic activity. Could Authors evaluate ATPase activity with an in-gel activity staining? Authors state that IF1 exerts this function under normal phosphorylating conditions, although this is not really the case because experiments provided in this manuscript are mostly performed in isolated mitochondria without the addition of any respiratory substrates and ADP/Pi.

8. Authors measured IF1 and ATP synthase particles using high-resolution gold-immunolabeling approach. The number of ATP synthase and IF1 particles seems pretty underestimated, could Author comment on this? Moreover, it is not clear how Authors conclude from this set of experiments that the sluggish ATP synthase localizes at the cristae tips considering that the colocalization with IF1 is higher along the sheets.

Minor points:

1. Figure 1, I believe that panel C is dispensable, since it is well established that IF1 is a mitochondrial protein that colocalizes not only with β subunit but also with all other mitochondrial markers.

2. Page 5, lines 101-102, Authors state that IF1 inhibits only a fraction of ATP synthase under normal physiological condition. I am not sure that you can refer to "normal physiological condition" since all the lines used here are established cancer cells.

3. Figure 1, panel G, there is a small band between the two major signals in the IC section, what is it? I would also suggest to adjust the title of the paragraph referring to the figure since data reported here cannot be translated to all cell types.

4. Authors further explored the correlation between IF1 and ATP synthase functions by analyzing many murine tissues expressing different levels of IF1. For the sake of consistency, I would suggest to provide BN images of skeletal muscle and brain (which lack in panel C of Figure 4), and of immunoprecipitates with the heart sample (which lack in panel E).

Reviewer #2 (Remarks to the Author):

This is a very interesting study by a great team.
Several issues need to be addressed.

Across the manuscript, there is a general lack of description of the analysis of the images. The authors state that the analysis was done "manually with ImageJ" which is not enough.

Across the manuscript, authors do not provide the number of times the experiment was repeated. There is no definition of n for each of the experiments. A biological replicate requires a repeat of the experiment using a different batch of cells or the same batch of cells run on a different date; otherwise, these are called duplicates and are not an "n". In any case, biological replicates cannot be the number of cells. However, in this manuscript it is not clear if the biological replicates are independent experiments.

Assembly for CV seems impaired not only in the oligomeric states but also CV_m, where the band detected in IF1KO migrates lower than the one in the IF1 expressing cells. The data suggest that IF1 affects CV assembly overall and not only their oligomeric states.

Based on the TEM results (Figure 2G), the cristae structure in the IF1KO is very different and can affect the observations mentioned above. Therefore, it is difficult to tell if reduced levels of CV are affecting the cristae architecture or vice versa.

Since the cristae in the IF1 KO cells is affected, have the authors looked at other main players in mitochondria cristae structure, such as MICOS complex or OPA1?

PLA is normalized by cells, but due to a smaller number of cristae and reduced area of mitochondria in KO cells the numbers may reflect it. Can the normalization be by mitochondria area?

Description of PLA analysis is lacking: figure 2F shows no CI+CIII in controls but are then shown to exist by BN (Fig2E). What is the reason for this discrepancy: This artifact could likely result from a permeabilization issue as the opened cristae in KO observed by EM, could have better access of antibodies to CI and CIII. Permeabilization in this study was done only with triton, which for big inner membrane complexes could not be properly detected in Ctrl cells. It is therefore suggested that an IFC with both cells and antibodies as a control for all the PLA done will address this artifact. Same for Fig. 2I, I would say an IFC in the same conditions for detection of γ -F1 is needed.

STED images are not really super-resolution, no cristae are observed.

Figure 3F: the 2 channels of the image appear to be shifted, resulting in green and red separation. This is an imaging artifact. The same X/Y shift can be observed in Fig3B. B-GFP should co-localize 100% with TMRE. If there is TMRE without B-GFP, this is an artifact.

The reader is left with some confusion as to the effects of IF1 on synthesis and hydrolysis. The increase in ATP synthesis (Figure 1D) and hydrolysis (Figure 1E) in IF1KO suggest that IF1 plays a role in both inhibition of synthesis and hydrolysis, and not as suggested in the introduction that IF1 only affects hydrolysis.

Blots in Figure 1G look peculiar, there are only two lanes labeled but in the bottom blots there are three samples; what is in the middle lane?

In Figure 2E, the authors claim that IF1 KO have more Super Complex. This is true for CI and CIV, but it is not the case for CIII, where less CIII is shown in the Super Complex. Since there is no Super Complex without CIII, how can that be explained?

Y axis in Figure 1E is mislabeled since the readout is absorbance at 340 and not relative light units (this is the readout used for ATP synthesis not hydrolysis).

Y axis in Figure 2C cannot read well.

Reviewer #3 (Remarks to the Author):

Review for the manuscript

"IF1 promotes oligomeric assemblies of sluggish ATP synthase and outlines the heterogeneity of the mitochondrial membrane potential"

In this work, Cuezva and his collaborators studied the effect of IF1-KO on ATP synthase assembly and functional consequences by using state of the art biochemical and microscopy approaches. Jose Cuezva is a well recognized expert in the field of IF1 affects on ATP synthase and physiology. From the data from their current study, he and his collaborators conclude that IF1 is required for ATP synthase oligomerization and at the same time say that it blocks ATP synthase activity. While the first conclusion is not brand new, but shown with several independent approaches here, the combination of the two conclusion is quite challenging and new. It would indeed suggest that rows of IF1-promoted dimeric ATP synthase as observed in several organism are inactive, which is an interesting but also provocative suggestion. It certainly adds a new perspective to the question, whether different subpopulations of ATP synthase exist in (the same) mitochondrion, and how this is linked to mitochondrial heterogeneity in term of proton motive force. To be ready for publication, however, some issues have to be fixed as described in the following sections.

Abstract:

Misleading statement: How can a tissue devoid of IF1 have a pool of ATP synthase with IF1-bound?

Minor: Is sluggish the correct term, sounds a bit in casual.

Introduction:

Mention additional references for the role of ATP synthase in cellular adaptation (in the current version: mainly self-citations)

Results

Altogether, the results would profit from a more detailed descriptions of the assays, the data and their significance. Single important results are often described very condensed.

Importantly, the model suggested here is that binding of IF1 is promoting oligomerization of ATP synthase on one hand but also its inactivation. Consequently, that would mean that rows of dimers of ATP synthase as observed in several organisms as shown by the Kühlbrandt group are rows of inactive (here: sluggish) ATP synthase. Since in these organisms no monomeric ATP synthase was detected (yet) by Cryo-Tomography, the question arises, where active ATP synthase is found and in which oligomerization state? Please comment on this conundrum. Also: what physiological meaning would have the inactivation of ATP synthase in respiratory tissues such as brain and heart? Liver, on the other hand, is highly glycolytic under resorptive conditions but has no IF1? Do you suggest backwards or forwards ATP synthase activity in the absence of IF1?

The tissue specific IF1 results are very interesting (Fig. 4D). However, please stay with hydrolase activity in the text, since this is what you have tested (page 10, line 222). To convince the reader that IF1 blocks ATP synthase and hydrolase the same way, an in vitro assay would be required. The ATP synthesis assay which was used here, tested for overall ATP levels (Fig. 1D). Please mention the luciferase test also in the main text, it is not clear from your description, how the ATP amount produced was calculated (right diagram), is this the slope of the kinetics or the end point (no hyperbolic course shown, though)? Do you refer to total cellular protein? If ATP production/time it must be unit per time. Since Fig. 1D is your central Figure for your claim that IF1 inhibits mitochondrial ATP synthesis this must be crystal clear. Show oligomycin effects and your calibration in the central figure, too. Also, is the mitochondrial mass changed in IF1-KO cells? Mitotracker green

staining would show this. If you have more mitochondria in IF1-KO, this must be normalized!

Fig. 1E: which test for hydrolysis? how is ATPase activity calculated from the time course? Here, it declines faster in IF1-KO cells. please comment, what you expect for isolated mitochondria in terms of coupling, are these conditions comparable to in cell mitochondrial ATP synthase activity.

Fig 1F: mitochondrial instead of nuclear staining would strengthen the statement of interaction within mitochondria. It is brave to conclude from the data that signals within 400 nm range indicate oligomeric assemblies, the resolution is not fine enough to show this.

Page 5 line 111: indicating inhibition of ATP synthase or hydrolase function?

Fig. 2A: was the cell number controlled after the assay or is this the number of seeded cells? IF1-KO could affect the growth rate, have you tested this? Please show. Also, is the mitochondrial mass changed in IF1-KO cells? Mitotracker green staining would show this. This is also important for the luciferase assay for ATP synthesis!

Fig. 2C: Check printed title at y-axis

Page 6 line 130: why is this so, what could be a possible explanation for increased SC formation? See Fig. 2F comment

Fig. 2E: explain the additional bands in CIII and CIV staining (CIII higher than SC; CIV lower than monomeric CIV. Why does single CI run in the same height as monomeric CV, although they have different molecular weights?

Fig. 2F: please demonstrate the localization of the PLA-dots within mitochondria, e.g. by MitoTrackerDeepRed staining. Could it be that the increased SC formation is due to reduced cristae surface which would force SC formation?

Fig. 3E,F: high resolution images of GFP-transfected cells with TMRE staining of MMP are shown: was this live cell STED microscopy? How did you avoid phototoxicity? Please give details on laser power. How was the line plot obtained, please draw the line into the specific single depicted mitochondrion.

Fig. 3G, H: please use the same Y-axis scale for better comparison.

MitoSox and TMRM have a different intramitochondrial localization, MitoSox is found in the matrix and TMRM in the IMM. The conclusion that the heterogeneity of MMP is partially due to IF1 is very interesting: do the authors suggest that IF1 has different binding affinity to ATP synthase within different regions of a single mitochondrion? How is this regulated/promoted?

Fig. 4 The results are very interesting and well presented.

Fig4G,I, check print of β in title of y-axis, what do the red and black arrowheads indicate? How do you differentiate between cristae tips and sheets?

Material and methods

Material and methods need more detailed descriptions.

CII- activity assay: Oxidation of DCPIP results in a product that absorbs at 600 nm?

TMRM: explain abbreviations, MS

TMRM: 100 nM used, this would result in quenching mode and the signal increases with lower MMP (lower TMRM uptake). Please explain or comment.

Imaging: SP8, equipped with which objective, which detectors, which laser. Please describe in detail. How exactly was the STED module used? Only the deconvolution software or also the STED imaging? Please provide information about the excitation and emission wavelengths.

Response to Reviewers

Reviewer #1 (Remarks to the Author):

The main conclusions of this manuscript are the existence of two ATP synthase pools that depend on IF1 binding, and the preferential location of IF1-dependent “sluggish” ATP synthase at the cristae tips.

As known from previous literature, IF1 is the canonical inhibitor of ATP synthase that limits ATP depletion under conditions of severe oxygen deprivation or the in presence of a dysfunctional respiratory chain, a specific function that is mainly modulated by acidic pH. However, new roles of IF1 in normoxic conditions are emerging to suggest that IF1 could also prevent ATP synthesis, although contrasting data have been reported so far. The role of IF1 in the oligomerization of ATP synthase is also controversial. The present work inserts in this scenario reporting that IF1 knock-out cells (cancer cell lines) show an increased rate of both ATP hydrolysis and synthesis, an increased basal respiration (which is explained by a higher amount of supercomplexes) and a reduction of ATP synthase oligomers. Authors also report that IF1 expression in different tissues correlates with the hydrolytic capacity of ATP synthase. Overall the text is fluent, the workflow is linear and data are clearly presented. Few typos are present (for example page 4, line 66 “affect” lacks the final “s”). I believe that the novelty of this work and the most interesting aspect is the hypothesis that IF1 promotes oligomerization of inactive ATP synthases within specific compartments (i.e. cristae tips) which correspond to hyperpolarized microdomains that would explain the heterogeneity of mitochondrial membrane potential. Nevertheless, I found that data provided in this manuscript do not entirely support this hypothesis, and I pointed out some issues that Authors should address before this work can be reconsidered for publication in Communications Biology.

Major points:

1. Figure 1, panel D, are the two cell lines equally sensitive to oligomycin? There is only one trace relative to the effect of oligomycin on ATP production and it is not clear to which cell line it refers.

Yes, both CRL and IF1-KO cells are equally sensitive to oligomycin. Originally, and for the sake of figure simplification, we included only one trace representing the average ATP production rate of CRL and IF1-KO cells treated with oligomycin (new Fig. 1c). Following the reviewer’s comment, new Fig. 1, panel c, now incorporates the corresponding traces for oligomycin-treated CRL and IF1-KO cells, independently. As expected, in both cell lines the synthesis of ATP in the presence of oligomycin is negligible. A statement in this regard has been included in Results Section in lines 102-105.

2. Figure 1, panel E, it would be better to show the trace of mitochondria treated with oligomycin from the beginning (or a control trace with no oligomycin added) because as presented the drop of ATP hydrolysis could be due to exhaustion of the regenerating system. Are data of ATPase activity normalized to values obtained in presence of oligomycin?

When we set up the ATPase assay in the lab years ago (Bioprotocols 2016), we made sure that the system was not exhausted of NADH, the substrate that is being measured in the PK-LDH coupled reactions. However, following the reviewer’s request, new Figure 1, panel d, now incorporates the traces of ATPase activity in mitochondria treated with oligomycin from the

beginning of the assay, confirming the above statement. Yes, data of ATPase activity are normalized by subtracting the slope of the traces in the presence of oligomycin from the corresponding ones in the absence of inhibitor. This point has been indicated in the Methods Section in lines 741-744.

3. Figure 2, panel A, I would suggest to provide values of basal and oligo-sensitive respiration that have been previously subtracted for the non-mitochondrial respiration, which clearly differs between the two genotypes. Why ablation of IF1 decreases membrane potential (panel C) and ROS production (panel D) needs to be addressed. If this is due to increased ATP production the difference should be abolished by the addition of oligomycin, and this needs to be tested. Further, the Authors should include cyclosporin H to inhibit MDR pumps to make sure that the TMRM loading is the same in the WT and KO cells.

Following the reviewer's request new Figure 2, panel a, now incorporates histograms of the values of basal and oligo-sensitive respiration (OSR) subtracted for non-mitochondrial respiration.

We have previously shown that ablation of IF1 in different cell lines (JBC 2010, Mol Cell 2012) and in primary neuronal cultures (Plos Biol 2021) decreases mitochondrial membrane potential and ROS production, most likely because the backflow of protons into mitochondria is no longer impeded by the inhibition exerted by IF1 on the ATP synthase. Consistently, and as stated by the reviewer, these differences are obliterated by treatment of the cells with oligomycin (see Fig. 2i in Plos Biol 2021). Anyway, following the reviewer's request, new panels c and d in new Figure 2 now incorporate the effect of oligomycin on mitochondrial membrane potential and ROS production to show that differences in these mitochondrial parameters are obliterated. See lines 137-139.

To comply with reviewer's request, we have determined TMRM staining of WT and KO cells in the presence of cyclosporin H and found no relevant differences when compared to values without CsH (new Fig. 2c), thus ruling out the implication of MDR pumps in the differences in TMRM staining between the two genotypes. See lines 134-137.

4. The Authors state that the activity of respiratory chain complexes is unaltered between the two genotypes and thus the difference in basal respiration might be due to specific rearrangement of supercomplexes (SC). In Figure 2, panel E, SC are increased in membranes blotted against Complex I and IV components, while for CIII the difference is not evident at all (the spot on the top appears more an aspecific signal, could Author provide a better image?). I would also suggest to add loading controls for all presented BN gels. It would be important to provide a quantification of SC present. In the same panel, it seems that IF1 comigrates mostly if not entirely with ATP synthase monomers, although following Authors' hypothesis it should bind to and stabilize oligomers. Could Authors discuss this aspect?

Following the reviewer's request, we have included a new blot for CIII in new Figure 2, panel e, to show clear evidence of the increase in SC as result of IF1 ablation in the cells. In addition, we have included VDAC blots as loading controls of BN-gels. Moreover, we have quantified the relative increase in SC containing complexes I, III and IV experienced in IF1-KO cells to illustrate SC rearrangement in these cells (see new histogram in Figure 2e). We have no doubt that

respiratory complexes are rearranged in HCT116 IF1-KO cells, what might contribute to the observed increase in respiration and in minimizing ROS production.

Yes, we agree with the last comment of the reviewer and the explanation for this observation might stem from different factors. In the first place, the signal of IF1 antibody might be affected by the accessibility of the antibody to assemblies of the enzyme. In addition, oligomers of the ATP synthase represent a relatively small fraction of total mitochondrial ATP synthase and hence, differences in signal might be reflecting the different proportion of ATP synthase present in oligomers and monomers. Moreover, it has been described that IF1 interacts also with OSCP to prevent cell death (Cell Death & Disease (2023) 14:54) and with assembly intermediates of the ATP synthase to prevent wasteful hydrolysis of ATP during the assembly of the enzyme (PNAS (2018) 115:2988-2993). Thus, it is possible that IF1 signals with monomeric ATP synthase could also result from any of these alternatives. However, when comparing the in-gel activity of ATP hydrolysis in CN-gels of IF1-KO cells (new Fig. 1e) and IF1-KO tissue extracts (new Fig. 4f) to their respective controls, we support that IF1 also binds and inhibits the monomeric ATP synthase, not only oligomeric assemblies. A comment in this regard has been included in the revised version of the manuscript (see lines 395-400).

5. In Figure 2, panel G, electron microscopy images show major alterations of IF1 KO mitochondria which appear smaller and with fewer cristae than control (which is probably the most evident feature). Could Authors measure the number of cristae per mitochondria in the two genotypes? How can this be reconciled with an increased amount of SC?

Following the reviewer's request, we have measured the number of cristae per mitochondrion and the number of cristae per mitochondrial area in the cell (new data in Figure 2, panel g). In agreement with the reviewer's observation mitochondria of IF1-KO cells show a reduced number of cristae (8.8 ± 0.7 vs 5.2 ± 0.6 cristae/mitochondrion for CRL and IF1-KO cells, respectively). However, when the number of cristae is normalized by the mitochondrial area of the cell, we do not observe a reduced content of cristae between the two cellular genotypes (see new Figure 2g), supporting that the total mitochondrial mass of the cell is not different. Hence, we support that there is an increase in SC to support the enhanced respiration of IF1-KO cells. These data have been incorporated and discussed in lines 169-172.

6. Figure 3, Authors investigated whether presence of IF1 (and its binding to ATP synthase) correlates with a hyperpolarization by expressing a fusion IF1-GFP protein and a GFP (with the pre-sequence of subunit β) as control. Why is the GFP signal in pre β GFP-expressing cells almost the opposite to that of TMRM (panel F)? This would imply that GFP locates in depolarized compartments, is this the case? Then, it would be better to use a mutant of IF1 (fused to GFP) that does not bind ATP synthase as control for this experiment because it would normalize the effect of IF1 overexpression (likely present with IF1-GFP construct). I am a bit concerned by the fact that the two cell lines have been selected with different antibiotics (geneticin versus bleomycin) and therefore they are not really comparable. Moreover, Authors should check whether the fusion protein (IF1-GFP) can still interact with ATP synthase, which is not obvious. This is an important piece of information that needs to be provided. In general, I think that these

findings do not demonstrate beyond reasonable doubt that IF1 localizes in hyperpolarized domains.

We partially agree with the reviewer that in our effort to better illustrate the differences in the mitochondrial distribution between IF1-GFP and GFP we selected an extreme situation in Figure 3f. No, GFP is not located in depolarized compartments. In fact, the GFP signal is evenly distributed in pre β GFP stained mitochondria. Anyway, and per reviewer's request, the STED experiments with TMRM have been redone incorporating the control requested and a new experimental approach. Briefly, we changed the depletion laser and STED microscope and the procedure to calculate the colocalization of TMRM and GFP staining using in this case whole mitochondrial staining of the cell rather than the distribution of fluorescence in the line defining the organelle axis. The new approach allows better visualization of discrete mitochondrial regions (new Figure 3) further supporting that IF1 colocalizes with hyperpolarized domains.

Following the reviewer's request, we have generated the IF1-S39E-GFP mutant (new Figure 3b and Supplemental Fig. 3a), a mutation that we have previously shown to prevent the binding of IF1 to ATP synthase, thus lacking its inhibitory activity on ATP synthetic and hydrolytic activities of the enzyme (Cell Rep. 2015). In this way, we also satisfy the concern raised by the reviewer regarding the differences in the antibiotic used for selection of the cells. Now, both IF1-GFP and IF1-S39E-GFP cell lines have been selected with bleomycin.

The results show (new Figure 3a-c, and Supplemental Fig. 3a) that the three constructs used in the study IF1-GFP, IF1-S39E-GFP and p β -GFP are efficiently targeted to mitochondria. Moreover, co-immunoprecipitation experiments indicate that none of the controls used, IF1-S39E-GFP or p β -GFP, interact with ATP synthase (new Figure 3d). Only the IF1-GFP construct interacts with the enzyme (Fig. 3d). STED microscopy (new Fig. 3e) and calculation of the co-distribution of IF1 with TMRM (Fig. 3f) clearly support that IF1 localizes within hyperpolarized domains of mitochondrial cristae regions when compared to the controls used. See lines 197-199; 203; 207-209; 210-224.

7. Figure 4, according to the Authors' hypothesis (IF1 binding to ATP synthase oligomers dampens enzymatic activity), I would expect that oligomers in BN gels should not show hydrolytic activity. Could Authors evaluate ATPase activity with an in-gel activity staining? Authors state that IF1 exerts this function under normal phosphorylating conditions, although this is not really the case because experiments provided in this manuscript are mostly performed in isolated mitochondria without the addition of any respiratory substrates and ADP/Pi.

To satisfy the reviewer's request, in the revised version of the manuscript we now show in CN-gels of HCT116 cells (new Fig. 1e) and mitochondria of mouse tissues (Fig. 4f) that the expression of IF1 dampens the activity of ATP hydrolysis, affecting both monomeric and oligomeric assemblies of the enzyme. These results are stressed in the revised version of the manuscript. See lines 107-109; 272-281; 395-400.

Finally, we agree in part with reviewer's comment. In the revised version of the paper we have included the assays of the ATP synthetic activity in isolated mitochondria of mouse tissues (requested under point 4 of R#3) (new Fig. 4d). These assays were carried out with mitochondria energized with succinate and including ADP/Pi, i.e., under phosphorylating conditions, which we assume are those prevailing also in permeabilized cells (Fig. 1c). To comply with reviewer's comment, we now refer to basal cellular conditions.

8. Authors measured IF1 and ATP synthase particles using high-resolution gold-immunolabeling approach. The number of ATP synthase and IF1 particles seems pretty underestimated, could Author comment on this? Moreover, it is not clear how Authors conclude from this set of experiments that the sluggish ATP synthase localizes at the cristae tips considering that the colocalization with IF1 is higher along the sheets.

Gold immunolabeling of ultrathin sections involves harsh conditions of fixation and embedding that greatly affect the accessibility and reactivity of the antibodies with the antigens. We agree with the reviewer that the labeling is not superb, especially for the IF1 antibody. However, both the labeling of IF1 and ATP synthase are very significant above background mimicking the relative expression of the proteins in mouse tissues as assessed by western blotting, hence they are reliable representing a first approach in the field.

Both proteins co-distribute in the mitochondrial compartments apparently being the labeling in sheets more abundant over than that in the tips. However, the length (μm) of the sheets of inner membrane in mitochondria represents by a much larger extent the length of cristae tips. Hence, when the labeling of these structures takes into consideration the μm of sheets and tips of cristae as viewed in EM images, the density of labeling at cristae tips is threefold higher than in sheets, in agreement with previous studies about the localization of ATP synthase in the inner mitochondrial membrane (see ref. 2, 38) and recent findings from other laboratories (ref. 40).

Minor points:

1. Figure 1, I believe that panel C is dispensable, since it is well established that IF1 is a mitochondrial protein that colocalizes not only with β subunit but also with all other mitochondrial markers.

We partially agree with reviewer's comment. However, following the advice of reviewer #3, we have maintained the panel in new Supplemental Fig. 1b to illustrate that both antibodies are specific and thus reliable for Proximity Ligation Assays. Specially, for IF1 antibody which was newly developed for this work.

2. Page 5, lines 101-102, Authors state that IF1 inhibits only a fraction of ATP synthase under normal physiological condition. I am not sure that you can refer to "normal physiological condition" since all the lines used here are established cancer cells.

We thank the reviewer for the suggestion. We have changed the text to acknowledge that the results described were obtained in cancer cells in culture. Now it reads: "supporting the idea that

a fraction of mitochondrial ATP synthase is inhibited by its binding to endogenous IF1 under basal cellular conditions. Similarly, the generation of Jurkat IF1-knockout cells (Supplemental Fig. 1c) also supported that IF1 is inhibiting a fraction of mitochondrial ATP synthase under basal conditions of cell culture (Supplemental Fig. 1d, e).

3. Figure 1, panel G, there is a small band between the two major signals in the IC section, what is it? I would also suggest to adjust the title of the paragraph referring to the figure since data reported here cannot be translated to all cell types.

It was a spillover from the CRL to an empty well placed between CRL and IF1-KO immunocaptures. To solve the problem, we have replaced the image in new Figure 1g by another biological replicate of the immunocaptures. Moreover, and as requested, we have changed the title of the paragraph to “HCT116 and Jurkat cells have a fraction of inactive ATP synthase”. See line 91.

4. Authors further explored the correlation between IF1 and ATP synthase functions by analyzing many murine tissues expressing different levels of IF1. For the sake of consistency, I would suggest to provide BN images of skeletal muscle and brain (which lack in panel C of Figure 4), and of immunoprecipitates with the heart sample (which lack in panel E).

We did not provide BN images of brain complex V in Figure 4 because they have already been reported in a recent publication from our group (PLoS Biol 2021). In that work it is shown that the content of IF1 in brain mitochondria of neuron specific IF1-knockout, transgenic IF1-overexpressing and control mice correlates with the content of oligomeric assemblies of Complex V. In any case, to satisfy the reviewer’s concerns we have included in the revised version of Fig. 4e images of BN gels for the brain and muscle as requested. Likewise, we have incorporated in new Figure 4g images of the immunoprecipitates of heart samples as requested. Interestingly, in heart samples both bands corresponding to phospho- and dephosphorylated IF1 co-precipitated with ATP synthase. Since IF1 might have three phosphorylated serine residues, and only phosphorylation of S39 prevents IF1 binding (ref. 28), we suggest that in heart there might be additional phosphorylated forms of IF1 that do not affect enzyme activity (ref. 28) and have an unknown relevance.

Reviewer #2 (Remarks to the Author):

This is a very interesting study by a great team.

We thank the reviewer for his/her appreciation of the study and of our group.

Several issues need to be addressed.

1. Across the manuscript, there is a general lack of description of the analysis of the images. The authors state that the analysis was done “manually with ImageJ” which is not enough.

We apologize for the lack of description. In the revised version, we have provided additional information in the text, figure legends and in Methods sections as requested. For instance, see lines 903-918, 931-939 and 981-994.

2. Across the manuscript, authors do not provide the number of times the experiment was repeated. There is no definition of n for each of the experiments. A biological replicate requires a repeat of the experiment using a different batch of cells or the same batch of cells run on a different date; otherwise, these are called duplicates and are not an “n”. In any case, biological replicates cannot be the number of cells. However, in this manuscript it is not clear if the biological replicates are independent experiments.

Once again, we apologize for the lack of clarity. In general, the experiments have been repeated at least three times in independent experiments carried out in three different days. In the Figure legends of the revised version of the manuscript we have clearly defined “n” for each of the experiments as requested either for cell lines and for the number of mice.

3. Assembly for CV seems impaired not only in the oligomeric states but also CV_m, where the band detected in IF1KO migrates lower than the one in the IF1 expressing cells. The data suggest that IF1 affects CV assembly overall and not only their oligomeric states.

We have no evidence to support that the lack of IF1 affects the assembly of monomeric complex V. In fact, it appears to be the opposite since its ablation increases both the synthase and hydrolase activities of the enzyme. However, in partial agreement with the reviewer’s observation, in the revised version of the manuscript we have replaced the blot of CV in Figure 2e because migration of the monomeric CV in that experiment was slightly anomalous, most likely because of the higher loading and/or because of the absence of co-migrating IF1. The abnormal mobility of monomeric Complex V has not been observed in other biological replicates of IF1-KO cells nor in IF1-KO mitochondria of different mouse tissues (Figure 4e), in agreement with previous findings in brain mitochondria of IF1-KO mice (see PLoS Biol 2021).

4. Based on the TEM results (Figure 2G), the cristae structure in the IF1KO is very different and can affect the observations mentioned above. Therefore, it is difficult to tell if reduced levels of CV are affecting the cristae architecture or vice versa.

Of course, and in partial agreement with reviewer’s comment, there are many factors that can influence mitochondrial cristae structure. We should stress that we are not observing reduced levels of CV (Supplemental Fig. 2b), only disappearance of oligomeric CV when IF1 is ablated.

It is well established that dimerization of CV bends the inner membrane locally, as a prerequisite for correct cristae formation at its tips. Moreover, recent findings revealed that IF1 promotes the interaction between dimers of ATP synthase to form inactive tetramers of the enzyme (Gu et al., 2019; Pinke et al., 2020). Furthermore, we have not observed any differences in the expression of other main players involved in cristae organization, such as OPA1 and MIC60 after ablation of IF1 (new Supplemental Fig. 2e, and see next point of the reviewer). Hence, and considering present knowledge of the main players involved in cristae organization, we support that absence of IF1 is responsible for preventing the oligomerization of ATP synthase in these cells and therefore for the observed alteration of cristae structure.

5. Since the cristae in the IF1 KO cells is affected, have the authors looked at other main players in mitochondria cristae structure, such as MICOS complex or OPA1?

Following the reviewer's request, we have studied by Western Blot the expression of two other players in mitochondria cristae architecture, such as OPA1 and MIC60 (new Supplemental Fig. 2e). We observed no differences in the expression of these proteins between the two genotypes. Therefore, these results support that the observed changes in cristae morphology most likely result from lack of ATP synthase oligomerization.

6. PLA is normalized by cells, but due to a smaller number of cristae and reduced area of mitochondria in KO cells the numbers may reflect it. Can the normalization be by mitochondria area?

PLA assays are heavily dependent on the affinity and specificity of the antibodies used. Hence, it is critical to document in PLA studies the appropriate labeling of mitochondria with the antibodies used (see new Supplemental Figs. 1b and 2c, f). In the literature, PLA data using antibodies against ATP synthase and/or IF1 have been presented as spots/area (Bou-Teen et al., Aging Cell 2022), spots/nucleus (Galber et al., Cell Death Disease 2023) or as percentage of spots/ μm^3 of mitochondria using a projection of PLA signals onto mitochondrial immunostaining of the cells with TOMM20 (Acín-Pérez et al., EMBO J 2023). Normalization by mitochondrial area is therefore possible but perhaps is asking too much to a technique that has certain limitations. It should be taken into consideration that PLA signals are heavily amplified and some of them are projected out of the mitochondrial reticulum.

As indicated under point 5 of R#1, when the number of cristae is normalized by the mitochondrial area of the cell, we do not observe a reduced content of cristae between the two cellular genotypes (see new Figure 2g), because although the average mitochondrial area is significantly diminished in IF1-KO cells, the overall mitochondrial area of the cell is not affected (see new Figure 2g). Moreover, calculation of mitochondrial volume of the cells using Mitotracker red staining and the Huygens software with object analyzer command revealed no significant differences between CRL (n=10) and IF1-KO (n=7) cells ($348 \pm 37 \mu\text{m}^3$ versus $341 \pm 24 \mu\text{m}^3$, respectively). Likewise,

the mitochondrial area of the cell as assessed by Mitotracker staining and/or by immunostaining with different mitochondrial proteins (γ -F1 and UQCRCII) showed no differences between CRL (n=37) and IF1-KO (n=34) cells ($79\pm 2 \mu\text{m}^2$ versus $78\pm 2 \mu\text{m}^2$, respectively). Since the area and volume of mitochondria of CRL and IF1-KO cells are not variables of this study, the results when PLA signals are expressed in dots/ μm^3 of mitochondrial volume reproduce the differences between CRL and IF1-KO cells when the signals are expressed as dots/cell: PLA IF1/ β -F1: 0.018 ± 0.002 versus 0.002 ± 0.0004 dots/ μm^3 of mitochondrial volume, for CRL and IF1-KO cells, respectively; PLA CI+CIII: 2.7 ± 0.2 versus 5.2 ± 0.4 dots/ μm^3 of mitochondrial volume, for CRL and IF1-KO cells, respectively; PLA γ -subunit: 4.5 ± 0.7 versus 0.2 ± 0.05 dots/ μm^3 of mitochondrial volume, for CRL and IF1-KO cells, respectively (data for reviewer's information). However, we believe that for comparison of two genetic backgrounds that show no differences in mitochondrial area or volume per cell, and once the appropriate PLA controls are presented, the best and less complicated way to present PLA results is in dots/cell, which better illustrate the results of the experiment. In any case, the results of PLA assays are supported by other experiments like blue-native gels and/or by co-immunoprecipitations. Anyway, trying to satisfy the reviewer's request, we have incorporated an image of the mitochondrial staining in each of PLA assays using Mitotracker red (new panels in Figures 1f, 2f and 2i), to show the localization of PLA signals projected onto mitochondrial staining of the cells. However, the calculation of PLA signals are expressed as dots/cell in the absence of Mitotracker staining because its inclusion significantly affected the efficiency of the polymerase. These points have been incorporated in the revised version of the paper (see lines 113-122).

7. Description of PLA analysis is lacking: figure 2F shows no CI+CIII in controls but are then shown to exist by BN (Fig2E). What is the reason for this discrepancy: This artifact could likely result from a permeabilization issue as the opened cristae in KO observed by EM, could have better access of antibodies to CI and CIII. Permeabilization in this study was done only with triton, which for big inner membrane complexes could not be properly detected in Ctrl cells. It is therefore suggested that an IFC with both cells and antibodies as a control for all the PLA done will address this artifact. Same for Fig. 2I, I would say an IFC in the same conditions for detection of γ -F1 is needed.

We agree with the reviewer's comment. The revised version of the manuscript accommodates description of PLA analysis and documents the appropriate controls of the immunoreactivity of the antibodies used in PLA towards the mitochondrial proteins that were not included in the original submission (new Supplemental Fig. 2c, f). The lack of CI+CIII in previous PLAs of control cells was due to the age of the batch of antibodies used. New Figure 2f now shows the presence of CI+CIII in controls and the sharp increase in CI+CIII in IF1-KO cells, in agreement with BN gels (Fig. 2e). We thank the reviewer for the suggestion of increasing the permeabilization of the cells. However, it was not necessary to increase permeabilization since the same conditions of IFC using a new batch of the antibodies worked nicely and much better than in previous PLA assays. Overall, the procedures of PLA assays and IFC are the same. New

Supplemental Fig. 2c, f, respectively provide the two controls requested by the reviewer for IFC of CI+CIII and γ -ATPase.

8. STED images are not really super-resolution, no cristae are observed.

Following the suggestion of R#1 in comment 6, we have developed new STED experiments of higher resolution. In the new STED images in New Fig. 3e, mitochondrial cristae can now be observed. In brief, the STED experiments with TMRM have been redone incorporating an additional control (IF1-S39E-GFP); we have changed the depletion laser (from a 770 nm to a 670 nm) and switched to another STED microscope. Moreover, we have also changed the procedure to calculate the colocalization of TMRM and GFP staining. In the present study, we have used whole mitochondrial staining of the cells rather than the distribution of fluorescence in the line hand-draw defining the organelle axis. The results show (new Figure 3a-c, and Supplemental Fig. 3a) that the three constructs used in the study IF1-GFP, IF1-S39E-GFP and β -GFP are efficiently targeted to mitochondria. Moreover, co-immunoprecipitation experiments indicated that none of the controls used, IF1-S39E-GFP and GFP, interact with ATP synthase (new Figure 3d). Only the IF1-GFP construct interacts with the enzyme (Fig. 3d). STED microscopy (new Fig. 3e) and calculation of the co-distribution of IF1 with TMRM (Fig. 3f) support that IF1 localizes within hyperpolarized domains of mitochondrial cristae regions when compared to any of the controls used. The new STED approach allows better visualization of discrete mitochondrial regions within cristae (new Figure 3e).

9. Figure 3F: the 2 channels of the image appear to be shifted, resulting in green and red separation. This is an imaging artifact. The same X/Y shift can be observed in Fig3B. B-GFP should co-localize 100% with TMRE. If there is TMRE without B-GFP, this is an artifact.

As requested by point 6 of R#1, and your comment 8 above, we decided to increase the resolution of our STED approach in TMRM analysis using the 660-depletion line (encouraged by Leica and present at CNB, a nearby institution), instead of the 710-depletion line (recommended, but not encouraged) that is present in our institution and used previously. Hence, Figure 3 is new, and it has been corrected for the shifting. To this purpose, chromatic aberration of each image was corrected individually with the full correction-iterative fit algorithm of the Huygens deconvolution software. This is now indicated in the revised version (see lines 913-915). Regarding Fig. 3b (now new Supplemental Fig 3a) it corresponds to the immunostaining of the GFP-expressing fixed cells with anti-COX1 antibody. The lack of 100% colocalization between GFP and COX1 is due to intrinsic limitations of the immunofluorescence technique, such as the antibody specificity and/or diffusion of mitochondrial GFP during the fixation. Moreover, the high Pearson correlation coefficients in Supplemental Fig. 3a indicate a good colocalization. In any case, microscope shifting has been checked using 1.0 μ m fluorescent blue/green/orange/dark red TetraSpeck Microspheres (T7282), and we observed no shifting between two colors.

10. The reader is left with some confusion as to the effects of IF1 on synthesis and hydrolysis. The increase in ATP synthesis (Figure 1D) and hydrolysis (Figure 1E) in IF1KO suggest that IF1 plays a role in both inhibition of synthesis and hydrolysis, and not as suggested in the introduction that IF1 only affects hydrolysis.

We apologize if we were not able to convey our message correctly. For many years, IF1 has been considered an inhibitor only of the hydrolase activity of the enzyme. In the words of Prof. Walker, a unidirectional inhibitor of ATP synthase, acting on the enzyme to inhibit its hydrolase activity only when the acidic environment of the matrix could promote IF1 activation, which is a situation that happens under mitochondrial stressful conditions (refs. 14, 15). However, and in contrast to the generally accepted idea that IF1 functions as a unidirectional inhibitor of the enzyme, our laboratory has provided evidence, both in cells in culture (refs. 16, 17, 21, 22) and in isolated mitochondria of conditional tissue-specific mouse models of loss and gain-of-function of IF1 (refs. 6, 23-27), that the mitochondrial content of IF1 affects both the synthetic and hydrolytic activities of the ATP synthase. Hence, one of the messages conveyed with our contribution is that IF1 is functioning under normal basal conditions as an *in vivo* inhibitor of both the ATP synthase and hydrolase activities of the mitochondrial ATP synthase. The IF1-bound enzyme perhaps represents the sluggish ATP synthase described years ago by Peter Mitchell in some mitochondrial preparations. In the revised version of the manuscript, we have emphasized this point. As pointed out by the general comment of R#1, “new roles of IF1 in normoxic conditions are emerging to suggest that IF1 could also prevent ATP synthesis”, and that is one of the relevant points in our contribution. We have rewritten the part of the introduction (see lines 65-69).

11. Blots in Figure 1G look peculiar, there are only two lanes labeled but in the bottom blots there are three samples; what is in the middle lane?

As already mentioned under point 2 of R#1, it was a spillover from the CRL to a neighboring empty well between CRL and IF1-KO immunocaptures. To solve the problem, we have replaced the image in new Figure 1h by another biological replicate of the same experiment.

12. In Figure 2E, the authors claim that IF1 KO have more Super Complex. This is true for CI and CIV, but it is not the case for CIII, where less CIII is shown in the Super Complex. Since there is no Super Complex without CIII, how can that be explained?

It was due to a wrong choice of the appropriate image. As already mentioned under point 4 of R#1, we have included a new blot for CIII in new Figure 2, panel e, to show clear evidence of the increase in SC because of IF1 ablation in the cells. In addition, we have included VDAC blots as for loading controls. Moreover, we have quantified the relative increase in complexes I, III and IV experienced in IF1-KO cells to illustrate SC rearrangement in this situation (see new histogram in Figure 2e). We have no doubt that respiratory complexes are rearranged in HCT116 IF1-KO cells, what might contribute to the observed increase in respiration and in the diminished ROS production.

13. Y axis in Figure 1E is mislabeled since the readout is absorbance at 340 and not relative light units (this is the readout used for ATP synthesis not hydrolysis).

We apologize for the mistake. In the revised version, the labeling of ATPase activity in Figure 1d has been corrected.

14. Y axis in Figure 2C cannot read well.

The labeling of $\Delta\Psi_m$ in Figure 2c has been corrected, now reads well.

Reviewer #3 (Remarks to the Author):

Review for the manuscript

“IF1 promotes oligomeric assemblies of sluggish ATP synthase and outlines the heterogeneity of the mitochondrial membrane potential”

In this work, Cuezva and his collaborators studied the effect of IF1-KO on ATP synthase assembly and functional consequences by using state of the art biochemical and microscopy approaches. Jose Cuezva is a well recognized expert in the field of IF1 affects on ATP synthase and physiology. From the data from their current study, he and his collaborators conclude that IF1 is required for ATP synthase oligomerization and at the same time say that it blocks ATP synthase activity. While the first conclusion is not brand new, but shown with several independent approaches here, the combination of the two conclusion is quite challenging and new. It would indeed suggest that rows of IF1-promoted dimeric ATP synthase as observed in several organism are inactive, which is an interesting but also provocative suggestion. It certainly adds a new perspective to the question, whether different subpopulations of ATP synthase exist in (the same) mitochondrion, and how this is linked to mitochondrial heterogeneity in term of proton motive force. To be ready for publication, however, some issues have to be fixed as described in the following sections.

Abstract:

Misleading statement: How can a tissue devoid of IF1 have a pool of ATP synthase with IF1-bound?

We agree and thank the reviewer for the correction. The text “devoid of IF1” has been eliminated from the sentence.

Minor: Is sluggish the correct term, sounds a bit in casual.

We agree with the reviewer that sluggish sounds unusual to qualify an enzyme. However, since it was the term used by Moyle and Mitchell in ref. 8, we think that it is more appropriate to maintain the same terminology, at least in some places, to pay tribute to the father of bioenergetics. On the other hand, sluggish sounds catchy.

Introduction:

Mention additional references for the role of ATP synthase in cellular adaptation (in the current version: mainly self-citations)

Following the reviewer's request, we have incorporated the following additional references from other authors, emphasizing the role of ATP synthase in intracellular signaling (see new refs. 7-9 in line 47).

Results

1. Altogether, the results would profit from a more detailed descriptions of the assays, the data and their significance. Single important results are often described very condensed.

Following the reviewer's request, in the revised version of the manuscript we have added more details to the description of the assays, as well as to the relevance and significance of the results obtained. In fact, the text in results section has increased from 2144 words to 2981 words in the present version.

2. Importantly, the model suggested here is that binding of IF1 is promoting oligomerization of ATP synthase on one hand but also its inactivation. Consequently, that would mean that rows of dimers of ATP synthase as observed in several organisms as shown by the Kühlbrandt group are rows of inactive (here: sluggish) ATP synthase. Since in these organisms no monomeric ATP synthase was detected (yet) by Cryo-Tomography, the question arises, where active ATP synthase is found and in which oligomerization state? Please comment on this conundrum.

This is a very relevant question, but we must keep in mind that IF1 is not expressed in mitochondria of all cellular types. Hence, we cannot generalize because ATP synthase in liver and skeletal muscle mitochondria, that lack IF1 expression, show no inhibited (sluggish) enzyme but do have mitochondrial cristae which are presumably formed by rows of dimers of the enzyme. ATP synthase dimers in these tissues should be less stable than in those containing IF1 because they are not observed in BN-gels and in CN-gels, with the exception of a small amount of oligomers detected in brain CN-gels.

Our findings using gold decoration of the immunolabeled β -F1-ATPase and IF1 support that the two proteins are co-distributed within mitochondrial cristae and preferentially located at the tips, although a relevant fraction of both proteins is also found at cristae sheets. However, we also show that the mitochondrial content of sluggish ATP synthase is highly variable in mitochondria of mouse tissues and only a low fraction of the total enzyme is bound to IF1. We support that this picture represents a static view of the situation because the interaction of ATP synthase with IF1 should be highly dynamic and regulated by changes in local pH, $\Delta\Psi_m$, Ca^{2+} and other ions (refs. 3, 10, 15, 32), including the phosphorylation of IF1 (refs. 28, 29) or of other ATP synthase subunits (refs. 51, 52), which are likely to mediate active/inactive state transitions of the enzyme. In other words, we support that active ATP synthase is found at cristae tips preferentially over the sheets in any oligomeric state of the enzyme if it is not bound by IF1. Formation of the tetrameric structures that include dimers of IF1, as described by previous authors (refs. 27, 31, 32), are

definitively inhibited ATP synthase molecules whose abundance is highly variable in mammalian tissues. In any case, and to add further complexity to the conundrum, we show that the monomeric enzyme comigrates with IF1 both in cells and mouse tissues (Figure 2e and 4e). Remarkably, the in-gel ATP hydrolase activity of the enzyme, both in cells and mouse tissues (new Figures 1e and 4f), sharply increases when IF1 is ablated, supporting that the monomeric enzyme could also be bound and inhibited by IF1, a finding that can be generalized to other oligomeric states of the enzyme.

3. Also: what physiological meaning would have the inactivation of ATP synthase in respiratory tissues such as brain and heart? Liver, on the other hand, is highly glycolytic under resorptive conditions but has no IF1? Do you suggest backwards or forwards ATP synthase activity in the absence of IF1?

The sluggish ATP synthase represents a reservoir of enzyme that could become activated to supply ATP upon an increase in cellular energy demand, as we have reported in heart mitochondria *in vivo* in response to β -adrenergic signaling (ref. 28). Moreover, the sluggish ATP synthase is also a relevant generator of mtROS by the ETC for signaling and adaptation of different tissue responses, as we have documented in tissue specific transgenic and IF1-KO mice (refs. 6, 25). A comment in this regard has been now incorporated in the revised version of the manuscript (see lines 357-360).

At present time, we favor the idea that when IF1 is ablated in mitochondria in tissues that contain IF1 a futile ATP synthesis/hydrolysis cycle will be installed. We do not think that in liver or in skeletal muscle mitochondria, that are tissues naturally devoid of IF1, this futile ATP synthesis/hydrolysis cycle is operative under basal conditions. Obviously, these postulates need further investigations.

4. The tissue specific IF1 results are very interesting (Fig. 4D). However, please stay with hydrolase activity in the text, since this is what you have tested (page 10, line 222). To convince the reader that IF1 blocks ATP synthase and hydrolase the same way, an in vitro assay would be required.

We agree with the reviewer and following the reviewer's suggestion, the revised version of the manuscript now incorporates new panel d in Figure 4, to illustrate that ablation of IF1 also affects the ATP synthase activity of the enzyme, something that is not observed in tissues devoid of IF1, such as in liver and skeletal muscle. Therefore, we maintain the text as in the previous version.

5. The ATP synthesis assay which was used here, tested for overall ATP levels (Fig. 1D). Please mention the luciferase test also in the main text, it is not clear from your description, how the ATP amount produced was calculated (right diagram), is this the slope of the kinetics or the end point (no hyperbolic course shown, though)? Do you refer to total cellular protein? If ATP production/time it must be unit per time. Since Fig. 1D is your central Figure for your claim that IF1 inhibits mitochondrial ATP synthesis this must be crystal clear. Show oligomycin effects and your calibration in the central figure, too. Also, is the mitochondrial mass changed in IF1-KO cells? Mitotracker green staining would show this. If you have more mitochondria in IF1-KO, this must be normalized!

We agree with the reviewer's critique and comment and we apologize for it. The methodology to determine ATP synthase activity in permeabilized cells is now more clearly described in the Methods section (see lines 709-721) and in the text alluding the determination of ATP synthase activity (see lines 99-100). The method uses luciferase/luciferin to assess the kinetics of ATP synthesis over time (plot to the left, new Fig. 1c) in permeabilized cells supplemented with succinate and ADP. The histograms (to the right in Fig. 1c), show the quantifications of the slopes of CRL and IF1-KO cells ($\text{nmol ATP produced} \times \text{min}^{-1} \times \text{mg}^{-1}$ cellular protein) of five independent replicates. Yes, it is total cellular protein of permeabilized cells, and the activity is expressed as $\text{nmol ATP} \times \text{min}^{-1} \times \text{mg}^{-1}$ cellular protein.

As already mentioned under comment 1 to R#1, both CRL and IF1-KO cells are equally sensitive to oligomycin. Originally, and for the sake of figure simplification, we included only one trace representing the mean of ATP production of CRL and IF1-KO cells treated with oligomycin. Following the reviewer's request, new Fig. 1c, now incorporates the corresponding traces for oligomycin-treated CRL and IF1-KO cells. As expected, the synthesis of ATP in the presence of oligomycin is negligible. Moreover, new Figure 1c now incorporates the ATP calibration curve, as requested.

As commented under point 5 of R#1 and point 6 of R#2, when the mitochondrial area or mass of the cells is assessed by EM, by Mitotracker or by different immunostainings, we do not observe differences between the two cellular genotypes see (new Figure 2g) and lines 116-119, 153-155 in Results Section. Although the average area of each mitochondrion is significantly diminished in IF1-KO cells, the overall mitochondrial area of the cell is not affected (see new Figure 2g). Hence, we think that there is no need to normalize because the mitochondrial mass in both genotypes is the same.

6. Fig. 1E: which test for hydrolysis? how is ATPase activity calculated from the time course? Here, it declines faster in IF1-KO cells. please comment, what you expect for isolated mitochondria in terms of coupling, are these conditions comparable to in cell mitochondrial ATP synthase activity.

The ATP hydrolytic activity of the ATP synthase was spectrophotometrically determined with a coupled-enzyme assay using pyruvate kinase and lactate dehydrogenase as previously described by Barrientos 2009 and implemented years ago in our lab (Bioprotocols 2016). In this assay, it is measured the reduction in absorbance at 340 nm of the reaction mix due to the oxidation of NADH into NAD⁺ using the molar extinction coefficient of NADH ($\epsilon = 6.22 \text{ mM}^{-1} \text{ cm}^{-1}$) and the Lambert-Beer equation, considering that one NADH molecule is oxidized per each ATP molecule hydrolyzed by the ATP synthase. The faster decline in IF1-KO mitochondria indicates that the oxidation of NADH is faster in this genetic background, and hence, that the hydrolytic activity of

ATP synthase is higher in IF1-KO than in mitochondria of controls. These details are provided in Methods Section in lines 734-744.

No, definitively no, they are not comparable. The hydrolase activity of ATP synthase is assayed in isolated broken mitochondria under best conditions for assaying the activity whereas the synthase activity is assayed in permeabilized cells and/or isolated mitochondria that should maintain the integrity of the inner membrane to produce ATP under phosphorylating conditions. They are not comparable and that's one of the reasons why the hydrolase activity usually exceeds the synthase activity.

7. Fig 1F: mitochondrial instead of nuclear staining would strengthen the statement of interaction within mitochondria. It is brave to conclude from the data that signals within 400 nm range indicate oligomeric assemblies, the resolution is not fine enough to show this.

As already discussed, under previous points of R#1 and R#2, PLA data using antibodies against ATP synthase and/or IF1 are presented as spots/area (Bou-Teen et al., Aging Cell 2022), spots/nucleus (Galber et al., Cell Death Disease 2023) or as percentage of spots/ μm^3 of mitochondria using a projection of PLA signals onto mitochondrial immunostaining of the cells with TOMM20 (Acín-Pérez et al., EMBO J 2023). Normalization by mitochondrial area or volume is therefore possible but perhaps is asking too much to a technique that has certain limitations. It should be taken into consideration that PLA signals are heavily amplified and some of them are projected out of the mitochondrial reticulum. Since the area and volume of mitochondria of CRL and IF1-KO cells are not variables in this study, the results when PLA signals are expressed in dots/ μm^3 of mitochondrial volume reproduce the differences between CRL and IF1-KO cells when the signals are expressed as dots/cell: PLA IF1/ β -F1: 0.018 ± 0.002 versus 0.002 ± 0.0004 dots/ μm^3 of mitochondrial volume, for CRL and IF1-KO cells, respectively; PLA CI+CIII: 2.7 ± 0.2 versus 5.2 ± 0.4 dots/ μm^3 of mitochondrial volume, for CRL and IF1-KO cells, respectively; PLA γ -subunit: 4.5 ± 0.7 versus 0.2 ± 0.05 dots/ μm^3 of mitochondrial volume, for CRL and IF1-KO cells, respectively (data for reviewer's information). However, we believe that for comparison of the two genetic backgrounds that show no differences in mitochondrial volume of the cell, and once the appropriate PLA controls are presented, the best and less complicated way to present PLA results is in dots/cell, which better illustrate the results of the experiment. In any case, PLA data are further supported by other experiments like blue-native gels and/or by co-immunoprecipitations. Anyway, trying to satisfy the reviewers' requests, we have incorporated an image of mitochondrial staining in each of the PLA assays using Mitotracker red (new panels in Figs. 1f, 2f and 2i), to show the localization of PLA signals projected onto the mitochondrial reticulum of the cells. However, calculations of PLA signals are expressed as dots/cell in the absence of Mitotracker staining because its inclusion significantly affected the

efficiency of the polymerase. These points have been incorporated in the revised version of the paper (see lines 113-122).

We agree with the reviewer and the alluded statement regarding 400 nm has been eliminated from the revised version.

8. Page 5 line 111: indicating inhibition of ATP synthase or hydrolase function?

Indicating that IF1 inhibits both the synthase and hydrolase activities of the enzyme as we have also determined the ATP synthetic activity of the enzyme as a result of the new experiments.

9. Fig. 2A: was the cell number controlled after the assay or is this the number of seeded cells? IF1-KO could affect the growth rate, have you tested this? Please show. Also, is the mitochondrial mass changed in IF1-KO cells? Mitotracker green staining would show this. This is also important for the luciferase assay for ATP synthesis!

Yes, the protein content of the wells was determined after running the Seahorse experiments (aprox. 15h after seeding) and we found no differences between CRL and IF1-KO cells, in agreement with the lack of differences in growth rates of these cells (see new data on Supplemental Fig. 1a). Data in Figure 2a is expressed per number of seeded cells. To satisfy the reviewer's request the revised version of the manuscript incorporates the growth rate of the cells in Supplemental Figure 1a (see lines 94-95). As discussed previously, assessment of mitochondrial mass by many different approaches (R#1, p5; R#2, p6) indicated that the two cellular genotypes contain the same amount of mitochondria, so there is no need to correct by this variable. However, this point has been clarified in the revised version of the paper (see lines 116-119).

10. Fig. 2C: Check printed title at y-axis

We have corrected the y-axis in the new Figure 2c.

11. Page 6 line 130: why is this so, what could be a possible explanation for increased SC formation? See Fig. 2F comment.

We are not sure how the lack of IF1 affects supercomplex formation. Perhaps, as suggested by the reviewer, the finding that cristae length is smaller in IF1-KO cells could favor SC formation due to reduced surface of cristae sheets, helping in this way the aggregation of respiratory complexes that result in SC formation. We thank the reviewer for the idea, which has been incorporated in the text (see lines 409-411).

12. Fig. 2E: explain the additional bands in CIII and CIV staining (CIII higher than SC; CIV lower than monomeric CIV. Why does single CI run in the same height as monomeric CV, although they have different molecular weights?

The alluded additional bands were unspecific labeling by the antibodies. We have changed the CIII image for another blot in Fig. 2e without unspecific signals and where the increase of SC in

IF1-KO mitochondria is more evident. The blot for CV has also been changed per comment 3 of R#2 and the blot of CIV has been substituted by another blot not showing unspecific labeling. Since we use precast small gels, the difference in migration between CI and CV is not greatly accurate although CI does migrate slightly higher than CV. In the new Figure 2e, that has been adjusted accordingly.

13. Fig. 2F: please demonstrate the localization of the PLA-dots within mitochondria, e.g. by MitoTrackerDeepRed staining. Could it be that the increased SC formation is due to reduced cristae surface which would force SC formation?

Per reviewer's request and as previously discussed (see R#2, p6), PLA images now show the projection of PLA dots onto the mitochondrial reticulum. Yes, as already mentioned under point 11, perhaps, IF1-KO cells could favor SC formation due to reduced surface of cristae sheets, helping in this way the aggregation of respiratory complexes that result in SC formation. We thank the reviewer for the idea which has been incorporated in the text (see lines 409-411).

14. Fig. 3E,F: high resolution images of GFP-transfected cells with TMRE staining of MMP are shown: was this live cell STED microscopy? How did you avoid phototoxicity? Please give details on laser power. How was the line plot obtained, please draw the line into the specific single depicted mitochondrion.

Yes, they were high resolution images of stably transfected cells expressing GFP-targeted to mitochondria and cells stained with TMRM in live cell STED microscopy. As a result of comment 6 of R#1, and point 8 of R#2, we have developed new STED experiments of higher resolution. The new images have better resolution (see new Figure 3e) and cristae are distinguished. To avoid phototoxicity, we eliminated phenol red and fetal bovine serum (FBS) from the incubation media of the cells, cells were maintained at fixed CO₂ and temperature and the fluorophores excited with the less possible power of the lasers. The details are the following: For high resolution live imaging a Leica SP8 LSM, fitted with STED module at CNB Advanced Light Microscopy Facility was used with an immersion objective HCX PL APO CS 100x NA 1.4. Pixel size and z-stacks step distances were determined with the Nyquist calculator (Scientific Volume Imaging), to obtain optimal conditions for deconvolution. For GFP acquisition, a HyD 2 detector was used (488/498-528 nm wavelengths for excitation/collection with a white laser) applying a 592 nm STED depletion laser. For TMRM acquisition, a HyD 5 detector was used (552/564-668 nm wavelengths for excitation/collection with a white laser) applying a 660 nm STED depletion laser. For MitoSOX acquisition, a HyD 5 detector was used (552/560-632 nm wavelengths for excitation/collection with a white laser) applying a 660 nm STED depletion laser using. Once the STED images were acquired, the raw.lif images were corrected individually with the full correction-iterative fit algorithm and deconvoluted using Huygens deconvolution software, originating the images shown in the Figure 3e and Supplemental Figure 3b. The analysis of fluorescence in the previous manuscript was done in a hand-drawn line through the center long

axis of the tubular part of the mitochondrion. In the new study, the procedure to calculate the colocalization of TMRM and GFP staining used whole mitochondrial staining of the cells rather than the distribution of fluorescence in the line defining the organelle axis. That is the reason why we do not to present a line in new Figure 3e and new Supplemental Fig. 3b, c. All this information is now detailed in the corresponding Methods Section (see lines 898-920).

15. Fig. 3G, H: please use the same Y-axis scale for better comparison.

New Figure 3f and Supplemental Fig. 3c now show data with the same Y-axis scale.

16. MitoSox and TMRM have a different intramitochondrial localization, MitoSox is found in the matrix and TMRM in the IMM. The conclusion that the heterogeneity of MMP is partially due to IF1 is very interesting: do the authors suggest that IF1 has different binding affinity to ATP synthase within different regions of a single mitochondrion? How is this regulated/promoted?

In some way yes. What our data supports is that cristae subdomains of higher $\Delta\Psi_m$ are regions where IF1 is bound to the ATP synthase. However, this is a static view because the interaction of IF1 with the enzyme should be highly dynamic and regulated by changes in local pH, $\Delta\Psi_m$, Ca^{2+} and other ions (refs. 3, 10, 15, 32), including the phosphorylation of IF1 (refs. 28, 29) or of other ATP synthase subunits (refs. 51, 52), which are likely to mediate the active/inactive state transitions of the ATP synthase.

17. Fig. 4 The results are very interesting and well presented. Fig4G,I, check print of β in title of y-axis, what do the red and black arrowheads indicate? How do you differentiate between cristae tips and sheets?

We thank the reviewer for the positive comment. The printing of β has been amended in the revised new Figure 5. Black arrowheads point to dimers or triplets of ATP synthase and red arrowheads to ATP synthase particles near IF1, likely representing IF1-bound enzyme. We apologize for the mistake; this information was not included in the figure legend. Sheets are considered the planar region of cristae starting from IBM and ending 30-40 nm before the tip, whereas 30-40 nm of length of cristae at either side of the tips, which show a characteristic curve or ridge of the membrane, correspond to the tips. This information has been incorporated in the revised version of the text (see lines 313-314).

Material and methods

Material and methods need more detailed descriptions.

The material and methods section has been significantly extended including more details of the procedures performed. The length has been increased from 3302 words in previous version to 4630 word in the revised version.

CII- activity assay: Oxidation of DCPIP results in a product that absorbs at 600 nm?

Yes. DCPIP is used as an artificial electron acceptor to assess Complex II activity. When oxidized, DCPIP is blue with a maximal absorption at 600 nm; when reduced, DCPIP is colorless and does not absorb at 600 nm. Its extinction coefficient is $\epsilon = 21 \text{ mM}^{-1} \text{ cm}^{-1}$ {Barrientos, 2002 #10482}.

TMRM: explain abbreviations, MS

In the methods section of the revised version of the text, we have incorporated the entire name of the compound: Tetramethylrhodamine methyl ester. MS, we have not found the MS abbreviation in the text. On the other hand, we have not implemented mass spectrometry in this study.

TMRM: 100 nM used, this would result in quenching mode and the signal increases with lower MMP (lower TMRM uptake). Please explain or comment.

We apologize for the mistake. The concentration of TMRM used in all our experiments was 50 nM. We have corrected our error in the revised version of the text. TMRM was titrated before performing the experiments. At 50 nM concentration we observed no quenching of the signal. In new Figure 2c it can be observed that treatment of the cells with compounds that increase the MMP, such as oligomycin and cyclosporin H, also increase TMRM fluorescence, indicating that we are using the probe in non-quenching mode.

Imaging: SP8, equipped with which objective, which detectors, which laser. Please describe in detail. How exactly was the STED module used? Only the deconvolution software or also the STED imaging? Please provide information about the excitation and emission wavelengths.

For high resolution live imaging a Leica SP8 LSM, fitted with STED module at CNB Advanced Light Microscopy Facility was used with an immersion objective HCX PL APO CS 100x NA 1.4. Pixel size and z-stacks step distances were determined with the Nyquist calculator (Scientific Volume Imaging) to obtain optimal conditions for deconvolution. For GFP acquisition, a HyD 2 detector was used (488/498-528 nm wavelengths for excitation/collection with a white laser) applying a 592 nm STED depletion laser. For TMRM acquisition, a HyD 5 detector was used (552/564-668 nm wavelengths for excitation/collection with a white laser) applying a 660 nm STED depletion laser. For MitoSOX acquisition, a HyD 5 detector was used (552/560-632 nm wavelengths for excitation/collection with a white laser) applying a 660 nm STED depletion laser. Once the STED images were acquired, the raw.lif images were corrected individually with the full correction-iterative fit algorithm and deconvoluted using Huygens deconvolution software, that are the images shown in the Figure 3e and Supplemental Figure 3b. We have incorporated this missing information in the methods section of the revised version of the manuscript (see lines 898-920).

Overall, we thank the reviewers for the helpful editorial reviews of our work.

Reviewers' comments:

Reviewer #1 (Remarks to the Author):

We would like to congratulate the Authors for having provided a thoughtful and thorough revision addressing each point of concern with experiments and clarifications. We think that the manuscript has been strengthened and that it has all it takes to become a classic. The handling of all critiques speaks very highly of the Authors and of their commitment to Science.

Michela Carraro and Paolo Bernardi

Reviewer #3 (Remarks to the Author):

Review 1 for the revised version of
"IF1 promotes oligomeric assemblies of sluggish ATP synthase and outlines the heterogeneity of the mitochondrial membrane potential"

General comments:

The authors have done impressive work to answer the open questions and requests. They have added new results that strengthen their proposal. The key physiological question what role IF1 plays in regulating ATP synthase became new facets and yet remains enigmatic. How is local IF1 binding inside the same mitochondrion controlled? Does IF1 completely inhibit rows of ATP synthase oligomers? Do the brain and heart that rely mostly on OXPHOS and have the highest levels of IF1 the lowest level of ATP synthesis and if so how do cells cope with this conundrum? To stimulate the discussion in the field, these data should be published.

I was also asked to look at the rebuttal to reviewer 2 comments:

Review #2 revision Cuezva

There are still some issues that should be fixed before publication.

Suppl. Fig. 2c shows not blots as said in the supplementary figure 2c

Fig. 3e/f: Still not sufficiently explained, how the quantification was exactly done: Colocalization analysis of TMRE and GFP as Pixel-by pixel analysis? In the response, it is said, that whole cell analysis was done not mito-analysis. Since this is a crucial experiment, a clear description is required. It says further that maximal signals were used for comparison (Fig.3e, M1 vs. M2). How was this exactly done. Please provide a step-by step description

Your comments on the alignment of the two channels is not quite clear, yet. You state, that fluorescent beads showed no drift (which microscope) but you did an alignment via Hyugens (which microscope)? Please clarify.

Suppl. 4b, the definition of ATP synthase localized at cristae sheets is ambiguous, since ATP synthase oligomers decorate the cristae rims all along [1]. Only, if you would see cristae from the side view, or have 3 D- immunostaining it would be absolutely clear. Therefore, it would be rather safe to distinguish between cristae, cristae junctions and IBM.

Fig. 2e: strange that only two bands are seen for CIII, not three: CI/CIII2/CIV; CIII2/CIV, CIII2. How do you explain that free complex I vanishing in IF1-KO? The description and interpretation of the results of Fig. 2e is still too short.

[1] K.M. Davies, M. Strauss, B. Daum, J.H. Kief, H.D. Osiewacz, A. Rycovska, V. Zickermann, W. Kuhlbrandt, Macromolecular organization of ATP synthase and complex I in whole mitochondria, Proc Natl Acad Sci U S A 108(34) (2011) 14121-6.

Answers to reviewers

Reviewer #1 (Remarks to the Author):

We would like to congratulate the Authors for having provided a thoughtful and thorough revision addressing each point of concern with experiments and clarifications. We think that the manuscript has been strengthened and that it has all it takes to become a classic. The handling of all critiques speaks very highly of the Authors and of their commitment to Science.

Michela Carraro and Paolo Bernardi

We thank Michela Carraro and Paolo Bernardi for their participation in improving and very positive evaluation of our contribution.

Reviewer #3 (Remarks to the Author):

Review 1 for the revised version of

“IF1 promotes oligomeric assemblies of sluggish ATP synthase and outlines the heterogeneity of the mitochondrial membrane potential”

General comments:

The authors have done impressive work to answer the open questions and requests. They have added new results that strengthen their proposal. The key physiological question what role IF1 plays in regulating ATP synthase became new facets and yet remains enigmatic. How is local IF1 binding inside the same mitochondrion controlled? Does IF1 completely inhibit rows of ATP synthase oligomers? Do the brain and heart that rely mostly on OXPHOS and have the highest levels of IF1 the lowest level of ATP synthesis and if so how do cells cope with this conundrum?

To stimulate the discussion in the field, these data should be published.

We thank reviewer 3 for the positive evaluation of our contribution

I was also asked to look at the rebuttal to reviewer 2 comments:

Review #2 revision Cuezva

There are still some issues that should be fixed before publication.

Suppl. Fig. 2c shows not blots as said in the supplementary figure 2c

This is a misunderstanding. We did not mention blots for Supplemental Fig. 2c. It shows the mitochondrial immunostaining of the double immunofluorescence using antibodies against NDUFS5 (Complex I) and UQCRCII (Complex III), that were the antibodies used in PLA studies to confirm the increase in SC formation found in IF1-KO cells by BN-gels. We have not modified the text or figure.

Fig. 3e/f: Still not sufficiently explained, how the quantification was exactly done: Colocalization analysis of TMRE and GFP as Pixel-by pixel analysis? In the response, it is said, that whole cell analysis was done not mito-analysis. Since this is a crucial experiment, a clear description is required. It says further that maximal signals were used for comparison (Fig.3e, M1 vs. M2). How was this exactly done. Please provide a step-by step description

Yes, the analysis was done pixel by pixel. Once images were acquired, they were aligned and deconvoluted via Hyugens software. Images were analyzed with the JACoP plugin of ImageJ (NIH) to calculate the Mander's colocalization coefficients comparing the staining of GFP and TMRM pixel by pixel in entire images that contained on average 17 mitochondria. The

only parameter of the JACoP plugin that we selected for the analysis was the threshold of the intensity of the staining for each fluorophore. In both cases, we choose a restrictive threshold selecting only the maximal fluorescence pixels of GFP and TMRM, which was the same for all the images. The M1 and M2 Manders' colocalization coefficients are provided by the JACoP plugin. In the revised version, we have incorporated details of this description both in the material and methods section and in the legend of Figure 3.

Your comments on the alignment of the two channels is not quite clear, yet. You state, that fluorescent beads showed no drift (which microscope) but you did an alignment via Hyugens (which microscope)? Please clarify.

We apologize for the lack of clarity in our answers to reviewer 2 (point 9) of the previous submission. Reviewer #2 mentioned shifting in two figures obtained with different microscopes: 1, Supplemental Fig3a (old panel 3b) and 2, Fig. 3e (old panel 3f).

1. Supplemental Fig3a (old panel 3b) corresponds to the immunostaining of GFP-expressing fixed cells with anti-COX1 antibody. These images are acquired with a standard confocal A1R+ microscope (Nikon). The shifting of this microscope was checked with 1.0 μm fluorescent blue/green/orange/dark red TetraSpeck Microspheres. The results obtained revealed no shifting between green and red channels (see Figure 1 for reviewer's information). For this reason, any image obtained with this microscope (PLAs and immunostainings) has not been corrected for shifting.

Figure 1 for reviewer's information. Representative immunofluorescence image illustrating the colocalization of green and red signals of 1.0 μm fluorescent blue/green/orange/dark red TetraSpeck Microspheres using our A1R+ microscope (Nikon) and a 60x oil objective.

The lack of 100% colocalization mentioned by reviewer #2 is due to intrinsic limitations of the immunofluorescence technique as revealed by the high Pearson's correlation coefficients (see Supplemental Fig3a).

2. Fig. 3e (old panel 3f). These are STED images of GFP and TMRM staining of live cells. These images have been acquired with a Leica SP8 LSM microscope fitted with a STED module. However, due to the fact that the cells are alive and mitochondria are highly dynamic organelles, the shift between the red and green channels in this microscope is mainly due to the intrinsic movement of the organelles. For solving the shift, STED images have been aligned via Hyugens software as described in materials and methods.

Suppl. 4b, the definition of ATP synthase localized at cristae sheets is ambiguous, since ATP synthase oligomers decorate the cristae rims all along [1]. Only, if you would see cristae from the side view, or have 3 D- immunostaining it would be absolutely clear. Therefore, it would be rather safe to distinguish between cristae, cristae junctions and IBM. [1] K.M. Davies, M. Strauss, B. Daum, J.H. Kief, H.D. Osiewacz, A. Rycovska, V. Zickermann, W. Kuhlbrandt, Macromolecular organization of

ATP synthase and complex I in whole mitochondria, Proc Natl Acad Sci U S A 108(34) (2011) 14121-6.

We agree with the reviewer's comment that the majority of ATP synthase is placed at cristae rims, which is what our data supports. In fact, our calculation of gold decoration of ATP synthase at cristae tips shows that the majority of ATP synthase is placed at cristae rims (> 3-fold, see Fig. 5b). We also agree with the reviewer that the gold labeling at the sheets could represent ATP synthase placed in sheets or at the rim of the sheet and hence, that we have underestimated the content of ATP synthase at cristae rims. Since by *in situ* live cell microscopy ATP synthase has been found both at cristae edges and in sheets (ref 51), we have decided to maintain the presentation of our data but stressing in the text that the content of ATP synthase in cristae rims is underestimated. In other words, there is a small fraction of ATP synthase in sheets in agreement with ref51. Moreover, we indicate in the legend of Fig 5b and Supplemental Fig. 4c the reason why the estimation of ATP synthase at cristae tips represents an underestimation of ATP synthase at cristae rims. We have also incorporated in the text the reference suggested by the reviewer.

Fig. 2e: strange that only two bands are seen for CIII, not three: CI/CIII₂/CIV; CIII₂/CIV, CIII₂. How do you explain that free complex I vanishing in IF1-KO? The description an interpretation of the results of Fig. 2e is still too short.

In this cancer cell line the amount of CIII₂/CIV is always much less than that of SC or CIII₂ (see Fig. 3 in Nuevo-Tapioles (2020) Nat Commun 11, 3606). In fact, the CIII₂/CIV complex can be faintly observed in the overexposed blot (included herein for reviewer's information).

Figure 2 for reviewer's information. The abundance of complex CIII₂/CIV is very low in HCT116 colon cancer cells. (see also Nat Commun (2020) 11, 3606). The two blots shown differ in 2 min exposure.

The vanishing of free complex I in IF1-KO mice is explained by its integration into the SC as revealed by the sharp increase in the band intensity in SC in BN-gels (Fig. 2e), by the 2-fold increase in PLA signals using antibodies against complex I and complex III (Fig. 2f) and by the increase in cellular respiration (Fig. 2a).

To comply with the reviewer's request, these information has been incorporated in the revised version of the manuscript to increase the description and interpretation of the results of Fig. 2e.

REVIEWERS' COMMENTS:

Reviewer #3 (Remarks to the Author):

We thank the authors for responding to the outstanding issues. The manuscript is very interesting and now carefully prepared. The study and its results will further stimulate the discussion on the role of IF1. We look forward to its publication.